# α-Synuclein fibril and synaptic vesicle interactions lead to vesicle destruction and increased lipid-associated fibril uptake into iPSC-derived neurons

Amberley D. Stephens [1✉], Ana Fernandez Villegas [1], Chyi Wei Chung [1,5], Oliver Vanderpoorten[1,6], Dorothea Pinotsi[2], Ioanna Mela[1,3], Edward Ward [1], Thomas M. McCoy[1], Robert Cubitt[4], Alexander F. Routh[1], Clemens F. Kaminski [1] & Gabriele S. Kaminski Schierle [1✉]

Monomeric alpha-synuclein (aSyn) is a well characterised protein that importantly binds to lipids. aSyn monomers assemble into amyloid fibrils which are localised to lipids and organelles in insoluble structures found in Parkinson's disease patient's brains. Previous work to address pathological aSyn-lipid interactions has focused on using synthetic lipid membranes, which lack the complexity of physiological lipid membranes. Here, we use physiological membranes in the form of synaptic vesicles (SV) isolated from rodent brain to demonstrate that lipid-associated aSyn fibrils are more easily taken up into iPSC-derived cortical i3Neurons. Lipid-associated aSyn fibril characterisation reveals that SV lipids are an integrated part of the fibrils and while their fibril morphology differs from aSyn fibrils alone, the core fibril structure remains the same, suggesting the lipids lead to the increase in fibril uptake. Furthermore, SV enhance the aggregation rate of aSyn, yet increasing the SV:aSyn ratio causes a reduction in aggregation propensity. We finally show that aSyn fibrils disintegrate SV, whereas aSyn monomers cause clustering of SV using small angle neutron scattering and high-resolution imaging. Disease burden on neurons may be impacted by an increased uptake of lipid-associated aSyn which could enhance stress and pathology, which in turn may have fatal consequences for neurons.

[1] Department of Chemical Engineering and Biotechnology, University of Cambridge, Cambridge, UK. [2] Scientific Center for Optical and Electron Microscopy, ETH Zürich, Zürich, Switzerland. [3] Department of Pharmacology, University of Cambridge, Cambridge, UK. [4] Institut Laue-Langevin, Grenoble, France. [5] Present address: Department of Physics, Kavli Institute for Nanoscience Discovery, University of Oxford, Oxford, UK. [6] Present address: Department of Physics and Technology, UiT The Arctic University of Norway, Tromsø, Norway. ✉email: asd.stephens@gmail.com; gsk20@cam.ac.uk

In Parkinson's disease (PD), and other synucleinopathies, such as dementia with Lewy bodies and multiple system atrophy, insoluble aggregates form inside cells containing an abundance of a pre-synaptic protein called alpha-synuclein (aSyn)[1]. Monomeric aSyn is a 14.4 kDa soluble protein comprised of three defined regions, the N-terminus residues 1–60, which has an overall positive charge of 3+ and mediates lipid membrane binding, the non-amyloid component (NAC) region, residues 61–95, which is highly hydrophobic and aggregation prone, and the C-terminus, residues 96–140, which is highly negatively charged with an overall charge of −12, binds to cations and, in the presence of calcium, to synaptic vesicles (SV)[2]. aSyn preferentially binds to curved membranes, turning from an intrinsically disordered protein into having an α-helical structure at the N-terminus, that varies in length dependent on the lipid membrane properties[3,4]. Although its physiological function is not clearly defined, monomeric aSyn has been shown to cluster SV[2,5,6] and to be involved in SV release and recycling[6–8], while aSyn-knockout models show dysregulation in SV trafficking and homeostasis[9,10].

In cellular studies and patient samples, the insoluble inclusions called Lewy Bodies and Lewy neurites do not only comprise of aSyn, but lipids, organelle membranes and other proteins[1,11–14], including up to 21% of proteins related to SV[15]. In models overexpressing aSyn, which lead to aSyn aggregation, a dysregulation of vesicle homeostasis, vesicle numbers and localisation transpires[16–18]. In particular, small aSyn aggregates at the pre-synapse have been strongly linked to synaptic pathology[19]. There appears to be a fine balance between the lipid-associated physiological function of aSyn and a lipid-associated pathological function, as aSyn in monomeric[2,20,21], oligomeric[22,23], and fibrillar[24,25] states have all been shown to interact with lipid membranes. There is continued debate as to whether lipid membranes act as nucleation points and enhance aSyn aggregation or whether they reduce aSyn aggregation propensity by sequestering it into a less aggregation-prone state. These questions are not easily answered because aSyn-lipid interactions are mostly governed by electrostatic interactions, and are influenced by lipid composition, buffer, pH, presence of ions and protein:lipid ratios[21,26–30], all of which are hard to reproduce in vitro.

Furthermore, the discovery of aSyn fibril polymorphs in different cell types and patient tissues may indicate that the cellular environment, including the cell's lipid composition, plays a role in determining aSyn's final structure which subsequently may influence disease outcome[31–36]. Differences in fibril polymorphs have also been observed between aSyn fibrils formed in the presence or absence of synthetic lipids. The presence of lipids was shown to induce lateral clustering of straight fibrils, while other fibrils displayed a helical morphology with an increased width compared to those observed in the absence of lipids[37–39]. It is, however, unclear if incubation with physiological membranes leads to similar fibril polymorphs and whether these lipid-associated fibrils have differing pathological characteristics.

Several studies have been undertaken to characterise the interaction of aSyn with synthetic lipid vesicles and to understand the influence of different lipids on aSyn structure and its aggregation propensity[26]. Yet, few studies were carried out using physiological membranes, organelles or SV, which contain a broader range of lipids and proteins. Here, we use SV isolated from rodent brains, as a model physiological membrane with native lipid-associated proteins and containing diverse lipids, to study the interaction between monomeric and fibrillar aSyn with membranes and the subsequent influence of these interactions on the uptake of aSyn species into induced pluripotent stem cells (iPSC) differentiated into cortical neurons (so called i[3]Neurons). We show here that i[3]Neurons more easily take up lipid-associated

aSyn fibrils compared to aSyn fibrils alone. Yet, monomeric aSyn is more easily taken up than lipid-associated monomeric aSyn. We characterised the differences between lipid-associated and protein-only aSyn fibrils with coherent anti-Stokes Raman scattering (CARS) and stimulated Raman scattering (SRS) microscopy and show that SV lipids are directly associated with the aSyn fibrils which leads to differences in the resulting aSyn fibril morphology by atomic force microscopy (AFM), yet the core fibril structure remains the same. In terms of aSyn aggregation rates, we observe that the presence of SV increases the propensity of aSyn to aggregate into fibrils, although a high SV:aSyn ratio reduces nucleation rates and increasing the aSyn concentration increases elongation rates. Focussing on SV using small angle neutron scattering (SANS) and super-resolution microscopy, we show that fibrillar aSyn leads to the rapid disintegration of the SV lipid bilayer, while monomeric aSyn leads to SV clustering during the same timeframe.

Thus, using physiologically relevant SV membranes, we were able to shed light on the potential normal and pathological role of the interaction between aSyn and lipids and how lipid-associated fibrillar structures are more readily taken into neuronal cells and how fibrillar aSyn can have a detrimental effect on physiological membrane structures. Understanding the interaction between aSyn and physiological membranes is highly important for finding ways to stabilise functional interactions or inhibit pathological membrane interactions.

## Results

**Fibrillar aSyn is taken up more readily by i[3]Neurons when exposed to physiological lipids.** Within the brain, aSyn can be released and taken up by neighbouring neurons, leading to the transmission of so called aSyn seeds[40,41]. aSyn has a very high propensity to interact with lipids through its N-terminus containing 11 imperfect repeats which become helical upon binding, similar to apolipoproteins[42]. Within insoluble inclusions, such as Lewy bodies, observed in PD patients, aSyn is found highly enriched along with lipids[12,13]. Since both monomeric and small fibrillar aSyn can be released into the extracellular space, and due to aSyn's high affinity to lipids, it is conceivable that aSyn is associated with lipid structures when released into the extracellular space and/or bind to extracellular lipids, such extracellular vesicles, before subsequently being taken up into neighbouring neurons. Yet, there has been little study in the uptake of lipid-associated aSyn. As aSyn is primarily enriched at the presynapse where it is closely associated with SV, we used SV isolated from rodent brains[43,44] (Supplementary Fig. 1), as a model lipid system in our study. We thus compared the level of uptake into i[3]Neurons of aSyn monomers in the presence or absence of SV and of fibrils formed in the presence or absence of SV. Each of the above-described samples was sonicated, and the sonicated fibrils were analysed by AFM. The average length for fibrils formed in the absence of SV was 72.9 ± 45.2 nm and with SV 63.8 ± 35.1 nm (Supplementary Fig. 2). We treated i[3]Neurons with 500 nM aSyn monomer and fibrils +/− SV for one hour, 10% of aSyn was labelled with ATTO647N to allow for imaging. The aSyn uptake was quantified using direct stochastic optical reconstruction microscopy (dSTORM) with a resolution of ~20 nm[45] (Fig. 1a). We observe an increase in uptake of fibrillar aSyn grown in the presence of SV compared to aSyn fibrils grown in the absence of SV (Fig. 1b.i). We also quantified the length and number of fibrillar structures inside the neurons and show that the fluorescent aSyn species have a similar average length (179.9 nm for F, 182.3 nm for F + SV); but twice as many fibrillar species are taken up into neurons when the fibrils were incubated with SV compared to without (F + SV = 7334, F = 3695)

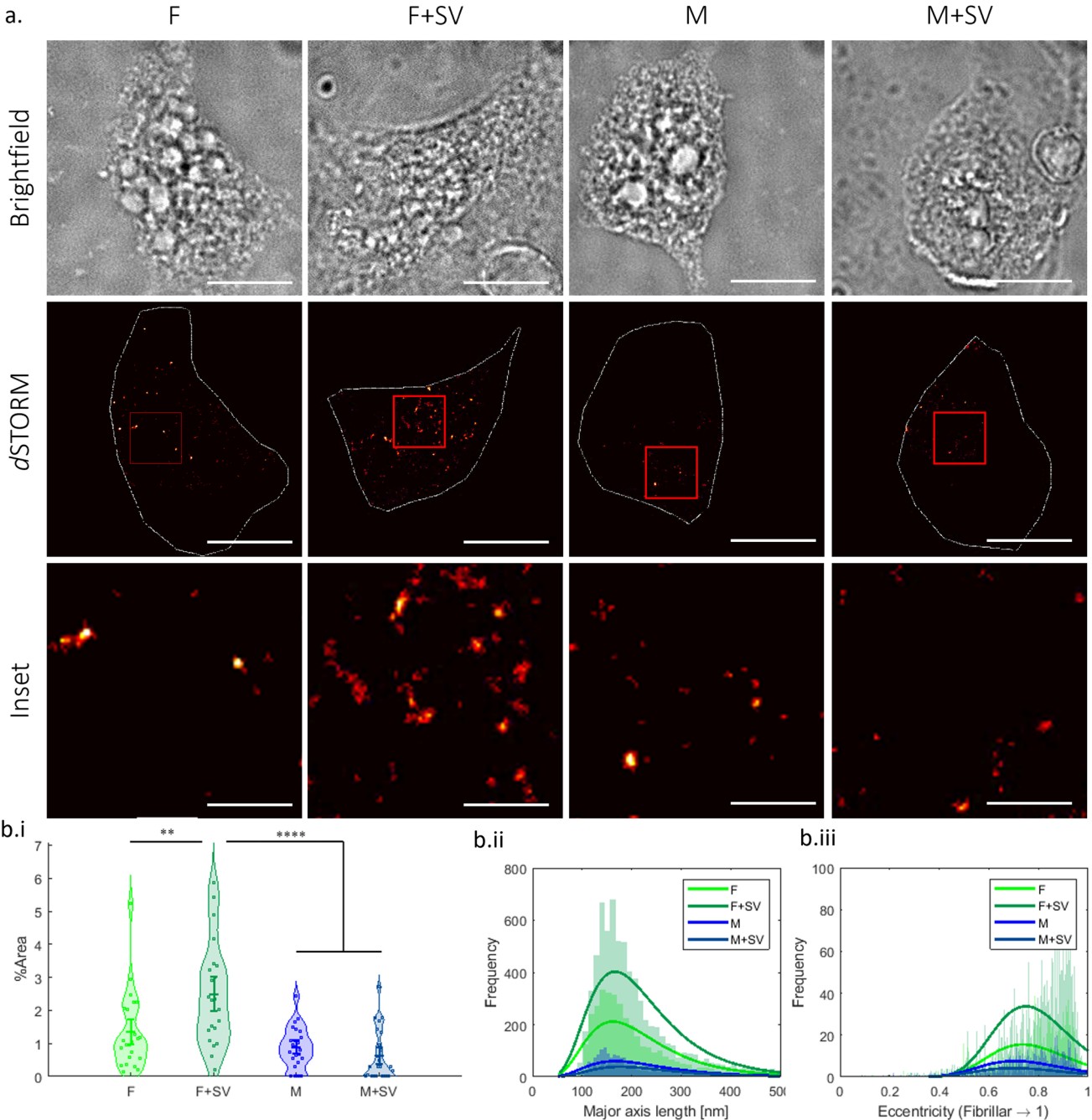

**Fig. 1 SV lipid-associated aSyn fibrils are more readily taken up into i³Neurons compared to aSyn only fibrils. a** 500 nM of 10% ATTO647N-labelled aSynC141 and 90% unlabelled WT aSyn as monomer (M) or fibrils (F) +/− SV were incubated with i³Neurons for 1 h before cells were washed, fixed and imaged. Shown are representative brightfield and fluorescence images of i³Neurons containing the different aSyn samples. The brightfield image was used as a reference point to define the edges of the cell soma for quantification of the soma area, highlighted by the white line in the *d*STORM images. Scale bar for brightfield and *d*STORM images = 10 µm. The insert highlighted by the red box in the *d*STORM images shows representative taken up aSyn structures in the soma, scale bar = 2 µm. **b**.i The taken up aSyn was quantified by the area of fluorescence divided by the area of the soma and is displayed as % area. There was a greater % area of fluorescence in cells that were incubated with sonicated fibrils grown in the presence of SVs (F + SV) compared to fibrils alone (F). There was no significant difference between monomer alone (M) and monomer with SV (M + SV), but less was taken up than for fibrillar aSyn samples. Experiments were repeated three times. A total of 23 cells were analysed per sample, each cell is represented as a point on the graph. A one-way ANOVA with Holm-Šídák tests was performed, F vs F + SV **$p < 0.005$, F + SV vs M and F + SV vs M + SV ****$p < 0.0001$. **b**.ii The quantity and major axis length show that aSyn fibrils grown in the presence of SV have an average major axis length of 179.9 nm and were more readily taken up by i³Neurons compared to fibrils formed alone which had an average length of 182.3 nm (F + SV = 7334, F = 3695). In contrast, more monomeric aSyn alone (M = 1962) was taken up compared to monomeric aSyn and SV (M + SV = 1040) which had average lengths of 187.1 nm and 195.4 nm, respectively. **b**.iii The eccentricity analysis shows that the fibril samples are more fibrillar in shape compared to monomeric aSyn, F = 0.823, F + SV = 0.831, M = 0.798, M + SV = 0.791. Dye only, SV only and dye with SV were used as controls (Supplementary Fig. 3).

(Fig. 1b.ii). In contrast, for monomeric only aSyn more fluorescent structures were observed compared to monomeric aSyn in the presence of SV (M = 1962, M + SV = 1040), while the size of the fluorescent structures was on average similar, M = 187.1 nm and M + SV = 195.4 nm (Fig. 1b.ii). As expected, measuring the eccentricity of the different aSyn structures reveals that aSyn fibril containing samples have a more fibrillar shape than monomeric aSyn containing samples, F = 0.823, F + SV = 0.831, M = 0.798, M + SV = 0.791, where a value closer to 0 is more circular and a value closer to 1 more ellipsoidal (Fig. 1b.iii). These data show that lipid-associated aSyn fibrils are taken up more readily into neuronal cells than non-lipid-associated aSyn fibrils, which may significantly influence disease burden.

**Addition of SV alters aSyn fibril morphology, but does not lead to aSyn fibril polymorphs.** We subsequently characterised differences between the aSyn only and the SV-lipid-associated fibrils and whether there were differences in fibril polymorphs compared to fibrils formed in the presence of synthetic lipid vesicles in the published literature. Coherent anti-Stokes Raman scattering (CARS) and stimulated Raman scattering (SRS) microscopy can be used to determine the chemical signature of samples. We therefore probed aSyn fibrils formed in the presence and absence of SV for a 'lipid signal'. We use a vibrational frequency at 2850 cm$^{-1}$ which corresponds to vibrations of $CH_2$ bonds, mostly found in lipids[46] and a frequency at 1675 cm$^{-1}$ which corresponds to the presence of protein due to characteristic stretching vibrations of $C=O$ bonds in amide I regions of β-sheet structures[47] (Fig. 2a.i). We normalised the signal intensity of the 'lipid signal' at 2850 cm$^{-1}$ to that of the signal at 1675 cm$^{-1}$ to account for differences in the quantity of sample in each field of view, and we observed that samples of aSyn aggregated with SV displayed a stronger lipid signal than those that had not been aggregated in the presence of SV (Fig. 2a.ii).

Growing evidence suggests that different polymorphs of aSyn can be formed in the presence of synthetic lipid vesicles comprised of different lipids and lipid to aSyn ratios[37,38]. Using AFM, we observe in the absence of SV, non-periodic, smooth aSyn fibrils (Fig. 2b.i). When aSyn monomer is incubated with SV (Fig. 2b.ii), the SV lipid-associated aSyn fibrils that are formed display two different morphologies, smooth fibrils (Fig. 2b.iii + iv, blue arrows) and periodic fibrils (Fig. 2b.iii + iv, pink arrows). After prolonged incubation, an increase in lateral clustering of fibrils occurs (Fig. 2b.v). Quantification of the fibrils shows that fibrils formed in the absence of SV have a height profile of 6.9 ± 2.6 nm (Fig. 2b.vi, blue), similar to fibrils previously measured[48]. Quantification of the SV lipid-associated fibrils shows that 80% of the fibrils are smooth, with a similar height to those formed in the absence of SVs, 6.7 ± 7.1 nm (Fig. 2b.vi, light pink), while 20% of the population of aSyn fibrils display a periodic morphology. Periodic fibrils have a peak height of ~12.1 ± 4.6 nm, a trough height of 7.1 ± 4.1 nm and a pitch distance of 175.2 ± 24.1 nm. The difference in peak height is almost double of the smooth fibril height, indicative of two fibrils twisting around each other (Fig. 2b.vi, dark pink). SV are not observed in the AFM images, likely due to their disintegration due to aSyn fibril formation.

Our results show that lipids from SV may be coating or intercalating into the fibrillar structure of aSyn. To address whether the SV lipids lead to changes in the core structure of the fibrils, we digested aSyn fibrils formed in the presence or absence of SV with proteinase K. Proteinase K can cleave residues that do not form part of the fibril core[32], and thus is able to indicate whether different fibril polymorphs form or not. We observe no difference in the proteolysis product profile of the aSyn only

fibrils or the SV lipid-associated fibrils (Fig. 2c, Supplementary Fig. 4). This suggests that SV lipids bind in between fibrils, leading to lateral bundling, rather than forming distinct aSyn fibril polymorphs as suggested previously using synthetic lipids.

**High SV:aSyn ratio reduces nucleation rates, while high aSyn concentration drives elongation rates.** We then investigated the influence of SV on aSyn aggregation rates and conversely the influence of aSyn monomer and fibrils on the structure of SVs in vitro. While many studies have investigated the effect of lipids on aSyn aggregation using different combinations of synthetic lipids, we use physiological SV containing a native lipid composition with SV-associated and transmembrane proteins incubated with 20 μM of aSyn to mimic concentrations of aSyn found at the presynapse[49]. We incubate samples with the small molecule thioflavin-T (ThT) which fluoresces when intercalated into β-sheet rich fibrillar structures and provides a fluorescence intensity-based read out to determine the aSyn aggregation kinetics. We compare the aggregation rate of 20 μM aSyn and 60 μM aSyn which mimics increased aSyn concentrations found in gene duplication and triplications[50–52]. As expected, we observe an increase in the aggregation rate of 60 μM aSyn compared to 20 μM (Fig. 3a), where the time for aSyn to nucleate and form fibrils, observed by the lag time ($t_{lag}$), decreases, and the aSyn elongation rate, defined by the slope of the exponential phase ($k$), increases (Fig. 3b, Table 1). When SV are added, the aggregation rates increase, which is also reflected in the remaining monomer concentration measured at the end of the kinetics assay, and by the maximum ThT fluorescence observed, where aSyn with SV have the least remaining monomer and increased maximum fluorescence compared to aSyn only (Table 1). However, increasing the ratio of SV to aSyn leads to an increase in the aSyn nucleation time, $t_{lag}$, with a ratio of 1:500 taking the longest to nucleate compared to lower ratios (Fig. 3b.i). The elongation rate is more dependent on the concentration of aSyn than on the ratio of SV to aSyn, where $k$ is consistently higher for reactions with 60 μM aSyn (shown by triangles in Fig. 3b.ii). $k$ reduces as the SV:aSyn ratio increases for 20 μM aSyn, but is similar for 60 μM aSyn (Fig. 3b.ii, Table 1). Although addition of SV increases aSyn aggregation rate in comparison to aSyn only, increasing the ratio of SV:aSyn ratio leads to a decrease in the nucleation and elongation rate of aSyn.

**SV cluster in the presence of monomeric aSyn, but become disintegrated by aSyn fibrils.** Finally, we investigated whether the presence of aggregated, fibrillar aSyn and monomeric aSyn can change the structure of the SV or can lead to SV disintegration. aSyn is known to disrupt synthetic lipid bilayers by distorting the membrane structure, inserting into the membrane, forming pores and/or nucleating on the membrane which can lead to the release of vesicle content[53–56]. Here, we use SANS and high-resolution imaging to investigate the integrity of the lipid bilayer of physiological SV in the presence of monomeric and fibrillar aSyn. The purified SV were imaged by TEM and ranged in sizes between 40 and 70 nm in diameter, a similar size to those isolated in ref. [43] and observed in situ by electron microscopy[57] (Supplementary Fig. 1). By using buffer solutions with 42% $D_2O$, to match the neutron scattering contrast of the protein with the bulk solvent, we could turn the protein on the SV and the aSyn in solution 'invisible' to neutron scattering. This allowed specific monitoring of the integrity of the lipid bilayer of the SV over time without convolution of the data from the protein signal (Fig. 4a). The SANS data were fitted using a Guinier-Porod model which provides the radius and a dimension variable (3-$s$) to define object shapes, where $s = 0$ represents spheres, $s = 1$ represents cylinders

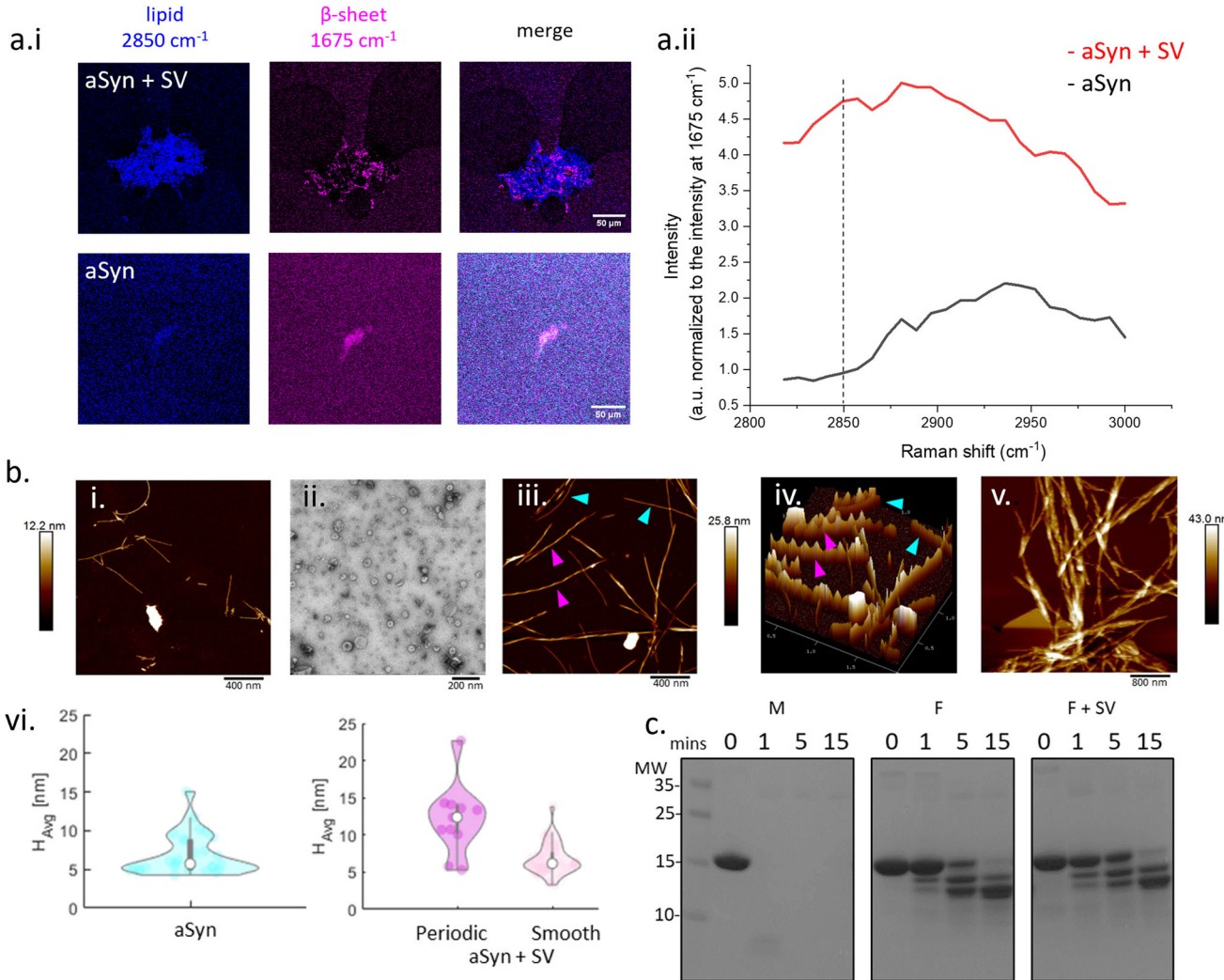

**Fig. 2 SV lipid-associated aSyn fibrils have a more periodic morphology than aSyn only fibrils, but do not form different fibril polymorphs. a**.i Lipid SRS signal at 2850 cm$^{-1}$ (blue) and the amide 1 region signal of a β-sheet structure at 1675 cm$^{-1}$ (pink), were acquired on top of the samples of aSyn fibrils grown with (aSyn + SV) and without SV (aSyn). Merging the signals at 2850 cm$^{-1}$ and 1675 cm$^{-1}$ reveals the lipid signal originating from the aSyn fibril cluster. **a**.ii Raman shift spectra of the aSyn and aSyn + SV samples show an increase in intensity of the lipid signal region for aSyn + SV compared to aSyn alone when normalised to the signal intensity at 1675 cm$^{-1}$. **b**.i Representative AFM image of aSyn fibrils grown without SV displays fibrils with a smooth morphology. **b**.ii A representative TEM image of SV added to aSyn monomer prior to incubation. **b**.iii 2D AFM image of aSyn incubated with SV showing fibrils with smooth (blue arrows) and periodic (pink arrows) morphologies. **b**.iv 3D AFM image of iii. more clearly shows the difference in periodicity of the smooth and periodic fibrils. **b**.v AFM imaging shows that aSyn fibrils and SV, both incubated for a week, lead to fibrils with increased lateral bundling. **b**.vi Analysis of the AFM images reveals that only smooth fibrils, with a height profile of ~6.9 ± 2.6 nm (n = 13 images), grow in the absence of SV (blue). 20% of fibrils grown in the presence of SVs (pink) were periodic and had a peak height of ~12.1 ± 4.6 nm, while smooth fibrils (light pink), comprising 80% of the sample, were on average 6.7 ± 7.1 nm in height (n = 25 images). **c** aSyn monomer (M), fibrils (F) and fibrils grown in the presence of SV (F + SV) were incubated with proteinase K for 0, 1, 5, and 15 min and the digestion products were separated on an SDS-PAGE gel, which was subsequently stained with Coomassie blue. The digestion profiles reveal no differences between different aSyn structures formed. Molecular weight (MW) markers are shown in kDa.

or rods and s = 2 represents platelet shapes[58] (Table 2, Supplementary Fig. 5). We collected data every hour over 45 h. At the end timepoint the signal intensity from the SV lipid bilayer in the presence of aSyn monomer increases compared to the SV only control and SV with aSyn monomer at 0 h (Fig. 4b.i). The radius of the SV lipid bilayer increases over time from 61.3 ± 3.8 nm to 97.9 ± 11.8 nm, indicating clumping or fusing of the SVs (Fig. 4b.ii, Table 2). These results complement our previous studies using fluorescence microscopy which also show that monomeric aSyn leads to clustering of SV[2,20], equally shown by others using vesicle mimetics[5,59] and in lamprey synapses[6]. Conversely, the lipid signal from SV incubated with preformed aSyn fibrils decreases rapidly (Fig. 4c.i). The loss of signal occurs

within 11 h and the data after 11 h could not be fitted due to the lack of form factor, but indicates total disintegration of the lipid bilayer over time (Fig. 4c.ii, d, 0–11 h grey-green). At time 0, the SV shape alters slightly in the presence of aSyn fibrils, indicated by the dimensionality factor, s, changing from 0.31 to 0.25, suggesting that the SV become more spherical as they begin to bind to the fibrils. The radius of the SV in the presence of both the aSyn fibrils and monomer at 0 h is similar to that in the absence of aSyn (59.4 ± 4.6 nm, 61.3 ± 3.8 nm, 70.1 ± 6.1 nm, respectively). The SV alone are still intact after 45 h, there is a slight, but insignificant change in the radius of the SV alone at 0 h compared to 45 h, while the changing dimension variable reveals that the SV become more spherical. The SANS data thus reveal the

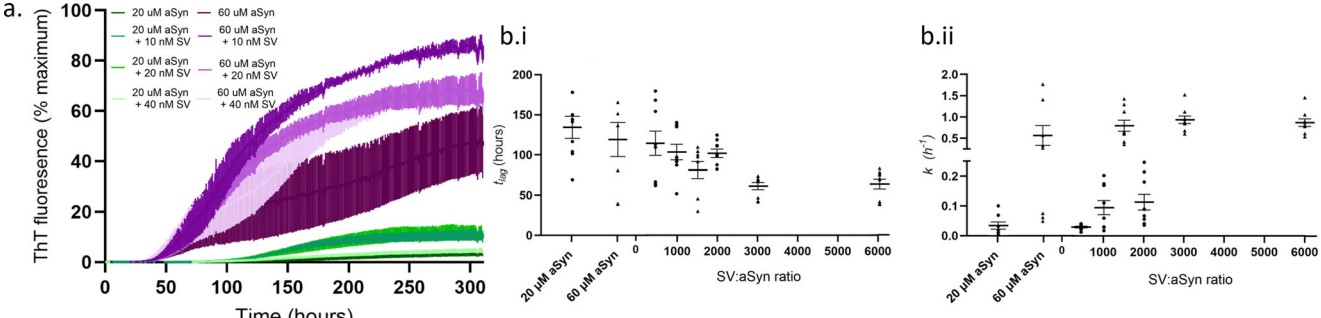

**Fig. 3 aSyn aggregation increases in the presence of SV, but decreases as SV:aSyn ratio increases. a** Aggregation rates of aSyn were observed by an increase in ThT fluorescence intensity, displayed as % of the maximum ThT intensity. An increase in concentration from 20 µM aSyn (green) to 60 µM aSyn (purple) leads to an increase in aSyn aggregation rate, while adding SV leads to an increase in the aSyn aggregation rate for all SV concentrations (10 nM, 20 nM, 40 nM) compared to no SV. However, addition of 40 nM of SV reduced the aSyn aggregation rate compared to addition of 20 nM and 10 nM SV. 60 µM ThT was added to each protein solution in a half area 96-well plate with orbital agitation at 200 rpm for 5 min before each read every hour for 310 h. Data represent 9 wells from 3 experimental repeats. **b**.i The aSyn nucleation rate, observed by the time to form fibrils, lag time ($t_{lag}$), was calculated and plotted against the ratio of SV to aSyn in nM concentrations. **b**.ii The aSyn elongation rate is determined by the slope ($k$) of the exponential phase plotted against the ratio of SV to aSyn in nM concentrations. Circles represent 20 µM aSyn, triangles represent 60 µM aSyn and error bars represent SEM.

**Table 1 aSyn concentration drives elongation, while SV affect nucleation kinetics of aSyn aggregation.**

| aSyn | SV | Ratio SV:aSyn | $k$ (h$^{-1}$) | $t_{lag}$ (h) | Remaining monomer (µM) | Maximum fluorescence |
|---|---|---|---|---|---|---|
| 20 µM | – | | 0.04 ± 0.01 | 134.5 ± 13.8 | 5.4 ± 0.4 | 3.9 ± 1.4 |
| | 10 nM | 1:2000 | 0.11 ± 0.03 | 101.3 ± 5.3 | 3.0 ± 0.0 | 10.7 ± 2.3 |
| | 20 nM | 1:1000 | 0.10 ± 0.02 | 103.6 ± 9.7 | 3.0 ± 0.1 | 11.5 ± 2.9 |
| | 40 nM | 1:500 | 0.03 ± 0.00 | 114.2 ± 15.0 | 3.2 ± 0.1 | 4.5 ± 1.0 |
| 60 µM | – | | 0.56 ± 0.03 | 118.7 ± 21.2 | 9.1 ± 1.4 | 48.8 ± 12.8 |
| | 10 nM | 1:6000 | 0.79 ± 0.13*** | 63.7 ± 6.1 | 3.4 ± 0.2 | 85.9 ± 3.4 |
| | 20 nM | 1:3000 | 0.80 ± 0.10** | 60.7 ± 4.4 | 3.3 ± 0.1 | 67.9 ± 6.5 |
| | 40 nM | 1:1500 | 0.91 ± 0.09* | 80.7 ± 10.4 | 3.6 ± 0.2 | 67.9 ± 3.3 |

Statistical tests were carried out using an ANOVA with Brown-Forsythe and Welch tests. Error represents s.e.m. Significance was detected for the elongation rate, $k$, comparing 20 µM to 60 µM aSyn rates, ***$p < 0.0003$, $p** < 0.0016$, $p* < 0.0069$, respectively.

detrimental effect preformed fibrillar aSyn has on the integrity of the SV lipid bilayer.

To visualise aSyn fibril and SV interactions in more detail, we used stimulated emission depletion (STED) microscopy. We observe an increase in the association of SV, labelled with a lipid-intercalating dye, mCLING, and preformed aSyn fibrils, 10% labelled with ATTO594, over 24 h which together form large mesh-like clumps (Fig. 4e.i, Supplementary Fig. 6). In both STED microscopy and AFM images, we observe some fibrils to be coated with SV while other fibrils and SV are not associated (Fig. 4e.i, Supplementary Fig. 7). After 24 h, the signal from the lipid intercalating dye, mCLING, is less punctate when observed in association with fibrils, indicating dispersion of SV over the fibril structure (Fig. 4e.ii white arrows, Supplementary Fig. 6). Higher resolution TEM images further confirm that there is an increase in the number of SV binding to fibrils over time (Supplementary Figure 8), but also reveal that smaller SV or broken SV formed into blebs adhere to the aSyn fibrils, supporting the above SANS results of the disintegration of SV (Fig. 4f). The time scale of SV disintegration in the SANS experiments is faster due to the higher concentration of aSyn and SV used to produce higher scattering counts, but also due to the presence of D₂O, which increases aSyn aggregation propensity[60]. We further show that, similar to our previous observations[2], incubation of monomeric aSyn with SV leads to the formation of small clusters after 24 h (Supplementary Fig. 9). When aSyn and SV are incubated separately as controls, there is no gross change

in morphology of aSyn monomer, preformed aSyn fibrils or SV (Supplementary Fig. 10). To ensure we image both fibrils and SV rather than artefacts, as some lipid intercalating dyes like mCLING form small micelles similar in size to SV, we use correlative STED-AFM. We observe fairly consistent fluorescence associated with SV, but also 'dark' fibrils are observed where no dye-labelled aSyn is incorporated into the fibrils, as seen previously (Supplementary Fig. 11)[61]. Our SANS and imaging data show fibrillar aSyn can lead to the disintegration of the SV, which in a cellular environment may be very damaging to the neuron.

## Discussion

aSyn is known as a lipid-binding protein, which is attributed to its physiological function in SV release and recycling, as well as binding to the plasma membrane and other organelles. Yet, aSyn also colocalises with lipids and organelles in Lewy Bodies, which are pathological insoluble aggregates found in synucleinopathy patient's brains[1,12–14]. We aimed to further our understanding of aSyn interaction with physiologically relevant membranes by looking at the interplay of the functional (monomeric) and pathological (fibrillar) form of aSyn and isolated SV from rodent brains, which are more complex than synthetic vesicles, and how this may influence the uptake of aSyn into neurons.

We initially investigated whether the presence of SV lipids could influence aSyn uptake into i³Neurons. The release and

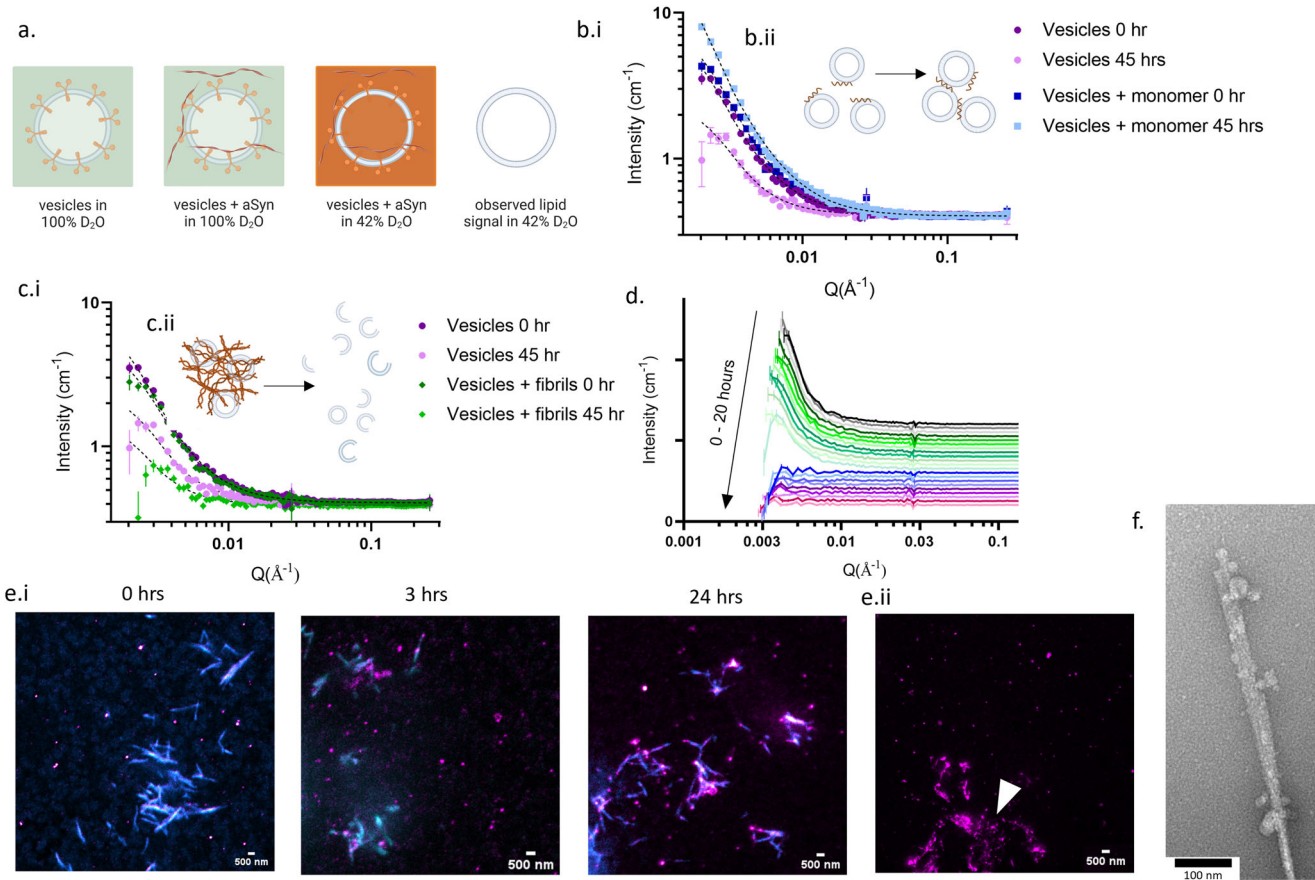

**Fig. 4 SV cluster in the presence of aSyn monomer, but degrade in the presence of aSyn fibrils. a** Schematic to represent SV proteins (orange) and lipids (blue) when in 100% $D_2O$ (green) and in the presence of preformed aSyn fibrils (orange), compared to in 42% $D_2O$ and 58% $H_2O$ (orange) allowing contrast matching of the protein, i.e. making all proteins 'invisible' to only observe the lipid bilayer of the SV. **b.**i SANS data of 1.5 mg/mL SV + 62 μM aSyn monomer (blue) compared to SV only (purple) at 0 h and 45 h, show an increase in signal intensity for SV + monomer from 0 h (darker colour) to 45 h (lighter colour). **b.**ii Schematic to indicate aSyn (brown) bound to SV (blue) leading to clustering of the SVs. **c.**i SANS data of 1.5 mg/mL SV only (purple) and with 50 μM preformed aSyn fibrils (green) show a decrease in signal intensity for SV + fibrils after 45 h. **c.**ii Schematic to indicate SV (blue) associating to preformed aSyn fibrils (brown) lead to the disintegration of the lipid bilayer of SV. In **b.**i and **c.**i, dashed lines show Guinier-Porod fitting and error bars represent s.d. (data for individual fits shown in Supplementary Fig. 5). **d** The signal intensity of neutron scattering from the lipid bilayer in the presence of aSyn fibrils over 20 h shows the loss of signal after 11 h (grey-green), no clear signal can be observed from 12–20 h (blue-pink), each line represents data collected each hour, data have been offset for clarity. Data for each condition were acquired once. **e.**i SV (0.5 mg/mL) were incubated with 5 μM preformed aSyn fibrils of 90% WT aSyn and 10% aSynC141:AF594 (cyan) and incubated at 37 °C for 0, 3, and 24 h. SV were stained with a lipid intercalating dye, mCLING:ATTO647N (1:100) (magenta). **e.**ii The mCLING fluorescence was less punctate and more spread over fibrils (indicated by white arrows) compared to the punctate mCLING signal in SV not associated to fibrils) (see Supplementary Fig. 6 for more images). **f** TEM image shows small blebs adhering to the fibril, indicating rupture and disintegration of the SV when associated with fibrils. Figure **a**, **b.**ii, **c.**ii made with Biorender.com.

uptake of aSyn is an important pathway for disease progression and may well involve lipid-bound aSyn structures. For instance, monomeric and oligomeric aSyn in and on exosomes can mediate cell to cell transmission[62,63]. We observe that aSyn fibrils are more abundant inside i3Neurons than monomeric aSyn, as seen by others adding exogenous aSyn to H4 neuroglioma cells and primary rat hippocampal neurons[64]. There are several pathways for aSyn internalisation, including endocytosis, receptor-mediated endocytosis, micropinocytosis and other, as-yet undefined pathways, which are dependent on the cell type and structure of aSyn[65,66]. Internalisation of fibrils, oligomers and monomer differ, where several receptors have higher affinity for aSyn fibrils compared to oligomers or monomers of aSyn, which include heparan sulfate proteoglycans[67], lymphocyte activation gene 3 (LAG3), amyloid precursor-like protein 1 (APLP1), and neurexin1β[68]. The uptake mechanism is proposed to occur via increased electrostatic interaction between receptors and fibrils compared to monomers due to a clustering of the negatively charged C-terminus in the fibrillar state. The increased uptake of

lipid-associated fibrils may be due to altered electrostatic interactions, or by enhanced uptake via endocytosis due to the presence of lipids, as observed with lipid micelle-mediated drug delivery[69]. Conversely, the opposite was observed for monomeric aSyn, as less aSyn monomer associated to SV-lipids is taken up into i3Neurons. While it has been shown that monomeric aSyn can diffuse across cell membranes, others have shown a greater uptake of cationic lipid-bound monomeric aSyn compared to monomeric aSyn only in primary rat hippocampal cells[70] and a HEK-293 stable cell line[71] via endocytosis, yet lipid-free aSyn may bind to the cell membrane surfaces more easily, leading to greater uptake compared to aSyn already associated to lipids. Further work is needed to determine the mechanistic differences in uptake of lipid-bound and lipid-free aSyn monomer in different cell types. The two-fold increase in uptake of lipid-associated fibrillar aSyn that we show is likely to increase stress to neurons as reported for the uptake of lipid-free aSyn fibrils[72].

We further characterised the SV lipid-bound fibrils to determine whether structural differences or simply the lipid-

**Table 2 Parameters of fitting SANS data presented in Fig. 4 obtained using the Guinier-Porod model.**

|  | Hours | Rg (nm) | Dimension variable (s) | Porod exponent |
|---|---|---|---|---|
| SV | 0 | 70.1 ± 6.1 | 0.31 | 1.95 |
|  | 45 | 59.9 ± 6.8 | 0.13 | 2.12 |
| SV + fibrils | 0 | 59.4 ± 4.6 | 0.25 | 1.94 |
|  | 45 | N/A | N/A | N/A |
| SV + monomer | 0 | 61.3 ± 3.8 | 0.32 | 2.00 |
|  | 45 | 97.9 ± 11.8 | 0.48 | 1.94 |

Error represents fitting error for residuals.

association resulted in the observed increase in aSyn uptake. The morphology of the resulting aSyn fibrils grown in the presence of SV was different to those grown without SV, displaying laterally bound fibrils, and increased fibril periodicity. Yet, we do not observe the tightly helical fibrils which have been reported to form in the presence of some small unilamellar vesicles (SUVs) made with negatively charged 1,2-didodecanoyl-sn-glycero-3-phospho-L-serine (DLPS), 1,2-dimyristoyl-sn-glycero-3-phospho-L-serine (DMPS) and dilauroyl phosphatidylglycerol (DLPG). It is worth noting though that the analysis techniques used in the previous studies differ to ours, and the highly helical fibrils were observed under dry conditions using TEM[37,38]. The CARS/SRS experiments revealed the presence of a strong lipid signal in aggregated aSyn samples grown in the presence of SV, suggesting association or intercalation of lipids into the fibril structures, as observed by others with synthetic lipid vesicles[24,39]. Our study showed that, although a twisted fibril morphology was present in the aSyn fibrils formed in the presence of SV, no aSyn polymorphs had formed as shown by proteinase K-based proteolysis analysis of the different aSyn structures. NMR and SAXS analysis of aSyn fibrils formed in the presence and absence of synthetic 1-palmitoyl-2-oleoyl-sn-glycero-3-phosphocholine (POPC) and 1-palmitoyl-2-oleoyl-sn-glycero-3-phospho-L-serine (POPS) SUVs show no difference in the fibril core structure[73,74]. Yet, recent cryoEM structures have shown different fibril polymorphs can be formed in the presence of POPC and 1-palmitoyl-2-oleoyl-sn-glycero-3-phosphate (POPA) SUVs, with lipids associated to residues 35–40 in one structures and 34–45 in another and intercalated between protofilaments[75]. The resolution provided by cryoEM combined with biochemical characterisation could provide clarity on whether fibrils formed in the presence of physiological SV or membranes have differing structure or contain lipids. Furthermore, a comparison is needed between fibrils formed in the presence of SV to those extracted from patients to determine whether it is possible to form fibrils in vitro that are relevant to disease-associated fibrils formed in vivo. However, another caveat to this comparison is whether the detergent-based extraction method of fibrils from patient tissues leads to the loss of associated lipids. Our data indicate that it is the lipid-association and not a difference in aSyn fibril structure that leads to the increased uptake into i3Neurons.

We further investigated the influence of physiological SV, displaying a complex lipid composition as mentioned previously[43,44], on aSyn aggregation kinetics. Studies using synthetic lipid vesicles have shown that the composition and lipid:protein ratio of the synthetic lipids greatly influence the aggregation propensity of aSyn[26,27], but only focus on two or three lipid mixtures at a time. Our studies reveal that the presence of SV increases the nucleation rate of aSyn compared to the absence of SV, likely in a similar mechanism to anionic synthetic vesicles, the SV can provide nucleation sites[76]. Yet, increasing the SV:aSyn ratio leads to a

decrease in the aSyn nucleation rate. The aSyn elongation rate also decreases as the SV:aSyn ratio increases yet is more influenced by aSyn concentration. A variety of negatively charged SUVs were shown to increase aSyn aggregation, but increasing the lipid ratio also increased the aggregation rate, while in the presence of neutral lipids no aSyn aggregation was observed[77]. SUVs formed of a mixture of neutral POPC and anionic palmitoyl-oleoyl-phosphoglycerol (POPG) decreased the aggregation rate of aSyn as lipid ratio increased due to aSyn having a higher affinity to the lipids than to the fibril ends[78]. As an IDP, aSyn is highly sensitive to its surrounding environment therefore the lipid composition of the SVs will greatly influence its aggregation propensity. Use of physiological membranes, salt buffers and pH are therefore highly important to understand aSyn aggregation propensity, a system with which we show that aSyn aggregation is significantly reduced.

Finally, we show that the interaction between preformed fibrillar aSyn and SV lead to the formation of large mesh-like structures, overtime the lipid bilayer of the SV disintegrates in the presence of aSyn fibrils. While, aSyn monomer initially leads to clustering of SV, as we have shown in the past[2,20], and by others using synthetic lipid vesicles[5]. Some fibrils appear decorated with SV, while others are bare. It has previously been hypothesised that different fibrils can lead to different lipid interactions due to electrostatic forces between the exposed amino acid residues and the different lipids present in different membranous structures. We do not, however, observe a distortion of the lipid membrane in the presence of fibrils by microscopy or SANS, as previously reported using synthetic SUVs formed of 1,2-dioleoyl-sn-glycero-3-phospho-L-serine (DOPS) and DLPS[79], and DOPS and 1,2-dioleoyl-sn-glycero-3-phosphocholine (DOPC)[24,39]. We instead observe fragmentation of SV into smaller vesicles/blebs by TEM. The lipids appear to spread along the fibril structures, as observed by the distribution of the lipid dye, mCLING, which, when combined with the reduction and complete loss of the scattering signal from the lipid bilayer, such as observed by SANS, suggest that aSyn fibrils can lead to a complete disintegration of the lipid bilayer of SV. Our data thus highlight the importance of studying aSyn-membrane interactions with physiological membranes which are more relevant to in vivo conditions and disease, as there are distinct differences in vesicle morphology and disintegration dependent on the choice of synthetic lipid in the SUV.

Incubation of SV with monomeric aSyn for 24 h leads to a shift in the shape of the SV as they become more ellipsoidal, most likely due to clustering of the SV as reported previously[2]. The integrity of the lipid bilayer, however, appears to be intact, which is opposite to what has been observed using synthetic lipid membranes upon incubation with monomeric aSyn over time, where leakage of contents occurs in SUVs and giant UVs (GUVs) formed of a variety of different lipid compositions[80,81]. This indicates that lipid composition and the presence of membrane proteins are crucial to stabilise SV in the presence of monomeric aSyn, which would indeed be important for aSyn to play a physiological role at the synapse. However, after prolonged incubation of SV and monomeric aSyn, disintegration of SV proceeds as aSyn fibrils form; indeed, no SV are observed by AFM after one week of incubation and the decoration of the lipid dye mCLING along the aSyn fibrils suggests vesicle disintegration and association of the vesicle lipids to the fibril surface. The extraction of lipids and destruction of SV could have dire consequences for dopaminergic neurons as the loss of SV not only leads to reduced neurotransmitter release at the active zone, but also to a reduced capacity for dopamine storage and thus to an increase of dopamine in the cytosol which significantly increases the production of reactive oxygen species[82].

To conclude we show that lipid-association rather than a change in fibril structure leads to the increase in uptake of aSyn

fibrils into i3Neurons. Disease burden may be impacted by the fact that lipid-associated aSyn is more easily taken up into neurons, which could seed endogenous aSyn. While release of lipid-bound and aggregated aSyn structures could also facilitate disease spread by increased uptake. We show how aSyn fibrils may enhance pathology by disintegrating SV, which may have fatal consequences, especially for dopaminergic neurons. If SV become disrupted or destroyed by aggregated aSyn they may release dopamine intracellularly which can lead to the formation of reactive oxygen species, damage to mitochondria and contribute to cell death[83]. Our study sheds light on the differences in experiments using synthetic lipid vesicles and our physiological SV, on resulting aSyn fibril polymorphs, vesicle shapes and disintegration. In the future further studies are needed into the cellular effects of lipid-associated aSyn structures and we may instead begin to assess therapeutics targeting toxic aSyn lipid-bound structures in the extracellular space or supporting functional lipid-bound aSyn to alleviate disease burden.

## Methods

**Purification of aSyn.** Human wild-type (WT) aSyn was expressed using plasmid pT7-7. The pT7-7 plasmid was also modified using site directed mutagenesis (#200523, QuikChange II, Agilent, Waldbronn, Germany) to incorporate a cysteine residue at position 141 to permit dye-labelling of the aSyn protein. The plasmids were heat shocked into *Escherichia coli* One Shot® BL21 STAR™ (DE3) (Invitrogen, Thermo Fisher Scientific, Cheshire, UK) and purified as previously described[84]. Recombinant aSyn was purified using ion exchange chromatography (IEX) in buffer A (10 mM Tris, 1 mM EDTA pH 8) against a linear gradient of buffer B (10 mM Tris, 1 mM EDTA, 0.15 M $(NH_4)_2SO_4$ pH 8) on a HiPrep Q FF 16/10 anion exchange column (GE Healthcare, Uppsala, Sweden). aSyn was then dialysed into buffer C (1 M $(NH_4)_2SO_4$, 50 mM Bis-Tris pH 7) and further purified on a HiPrep Phenyl FF 16/10 (High Sub) hydrophobic interaction chromatography (HIC) column (GE Healthcare) and eluted against buffer D (50 mM Bis-Tris pH 7). Purification was performed on an ÄKTA Pure (GE Healthcare). aSyn was concentrated using 10 k MWCO amicon centrifugal filtration devices (Merck KGaA, Darmstadt, Germany) and further purified to obtain monomeric aSyn using gel filtration on a HiLoad 16/60 75 pg Superdex column in 20 mM Tris pH 7.2 and stored at −80 °C until use. Protein concentration was determined from the absorbance measurement at 280 nm on a Nanovue spectrometer using the extinction coefficient of aSyn of 5960 $M^{-1}$ $cm^{-1}$.

Protein purity was analysed using analytical reversed phase chromatography (aRP). Each purification batch was analysed using a Discovery BIO Wide Pore C18 column, 15 cm × 4.6 mm, 5 μm, column with a guard cartridge (Supelco by Sigma-Aldrich, St. Louis, MO, USA) with a gradient of 95% to 5% $H_2O$ + 0.1% acetic acid and acetonitrile + 0.1% acetic acid at a flow-rate of 0.8 mL/min. The elution profile was monitored by UV absorption at 220 nm and 280 nm on an Agilent 1260 Infinity HPLC system (Agilent Technologies LDA, Santa Clara, USA) equipped with an autosampler and a diode-array detector. Protein purity fell ~95% dependent on batch.

**Dye-labelling of aSynC141.** Purification of aSynC141 was performed as above for IEX and HIC steps, but with the addition of 1 mM TCEP in all buffers to maintain the cysteine residue in a reduced form. After HIC, the protein was dialysed against 20 mM Tris 1 mM TCEP pH 7.2. The protein was then incubated with Alexa Fluor™ 594 C5 Maleimide (AF594) or ATTO647N (ATTO647) at a ratio of 4:1 dye to protein at 4 °C overnight. The protein was separated from free dye using gel filtration on a Superdex 75 pg 10/300 column (GE Healthcare).

**Purification of synaptic vesicles from rat brain.** Isolation of SV from rat brains was performed as in ref. [85] and detailed below. Tips for pouring the column, troubleshooting column issues and cleaning protocols can be found at ioprotocols.com dx.doi.org/10.17504/protocols.io.4r3l28zzjl1y/v1. Brains from two euthanized Sprague-Dawley rats were removed and washed in ice-cold homogenising buffer (320 mM sucrose, 4 mM HEPES, EDTA free c0mplete protease inhibitor, pH 7.4). The brains were homogenised in 9 ml homogenising buffer using a glass-Teflon homogeniser for 10 strokes at 900 rpm. The mixture was centrifuged at 1000 × g for 10 min at 4 °C. The supernatant was collected and centrifuged at 15k × g for 15 min at 4 °C. The supernatant was stored on ice and the pellet resuspended in 1 mL homogenising buffer. 9 mL of ice-cold ddH₂O with EDTA free complete protease inhibitors was added to the resuspended pellet and homogenised for three strokes at 2000 rpm. 50 μL of 1 M HEPES(NaOH) was immediately added after homogenising. The homogenate was centrifuged at 17k × g for 15 min at 4 °C. The resulting supernatant was combined with the supernatant on ice. The combined supernatants were centrifuged at 48k × g for 25 min at 4 °C. The supernatant was collected and homogenised for five strokes at

2000 rpm, drawn up and dispersed through a 30 gauge needle to disperse SV clusters. The supernatant containing SV was divided and 5 mL layered over a 5 mL 0.7 M sucrose cushion. The sucrose cushions were centrifuged at 133k × g for 1 h at 4 °C. 500 μL fractions from the cushions were removed starting at the top of the gradient. Fractions 12–20 were pooled and centrifuged at 300k × g for 2 h. The SV pellet was resuspended in 1 mL column buffer (100 mM Tris-HCl pH 7.6, 100 mM KCl) and homogenised in a 1 mL glass-teflon homogeniser for ten strokes at 900 rpm. The SV were drawn up and expelled through a 30 gauge needle three times. The SV were loaded onto a pre-prepared Sephacryl S-1000 column, and a peristaltic pump with a flow rate of ~6 mL h⁻¹ was set up in a cold room, 4 °C. Fractions of 0.7 mL were collected and analysed on a nanodrop to determine the presence of eluted SV by measuring the absorbance at 280 nm. The SV resided in the second peak shown on the chromatograph. The SV were pooled and centrifuged at 300k × g for 2 h, the pellet was resuspended in 100 mM Tris-HCl, 100 mM KCl pH 7.2 and aliquots were snap frozen. The final protein concentration of the SV was measured using a nanodrop spectrometer in μg/μL of protein.

**Animals.** Sprague-Dawley rats were bred and supplied by Charles River UK Ltd., Scientific, Breeding and Supplying Establishment, registered under Animals (Scientific Procedures) Act 1986, and AAALAC International accredited. All animal work conformed to guidelines of animal husbandry as provided by the UK Home Office. Animals were sacrificed under schedule 1; procedures that do not require specific Home Office approval. Animal work was approved by the NACWO and University of Cambridge Ethics Board.

**Small-angle neutron scattering.** Sample preparation: Monomeric aSyn was incubated in 20 mM Tris, 140 mM KCl, 0.05% NaN₃, pH 7.2 for 2 weeks at 30 °C to form fibrils. Fibrils and SV were dialysed overnight at 4 °C in 100 mM HEPES, 140 mM KCl in 42% $D_2O$ and 58% $H_2O$ in slide-a-lyzer™cassettes with 10KDa MWCO. Monomeric aSyn was buffer exchanged using PD10 midi traps (GE Healthcare) into the same buffers. 200 μL of SV were incubated with 200 μL of protein to reach a final concentration of 1.5 mg/mL SV, 50 μM aSyn fibrils, and 62 μM monomer.

Small angle neutron scattering (SANS) experiments [DOI: 10.5291/ILL-DATA.8-03-999] were carried out on D33[86] at the Institut Laue-Langevin – The European Neutron Source, Grenoble, France. The instrument was operated in monochromatic mode at a wavelength of 8 Å (full width half maximum 10%), with the rear detector placed at 12.8 m (centred on the direct beam, symmetrically collimated at a distance of 12.8 m) and the 4 front panels at 1.7–1.9 m, with various offsets compared to the direct beam to span as large a contiguous solid angle as possible. This configuration permitted to cover at once a broad q-range of 0.002–0.257 Å⁻¹ required to follow the kinetics. Samples were poured in quartz cuvettes of 1 mm pathway (120-QS, Hellma GmbH, Müllheim, Germany), installed on a temperature-controlled (37 °C) tumbling rack to constantly spin the cuvettes during the entire experiment, ensuring that the illuminated volume stays representative of the full content of the cuvettes despite possible phase separation. Each acquisition lasted 15 min, and 2 successive acquisitions of a sample were usually 15 min apart (except on occasions where controls were also measured). The program Grasp version 8.20b (Charles Dewhurst, ILL) was used to reduce the data, accounting for detectors parallaxes, flat fields and efficiencies, normalizing by acquisition time (flux is constant), using transmissions (measured at the beginning and the end of the kinetics), subtracting the contribution from an empty cuvette, and obtaining the absolute scale from the attenuated direct beam measurement, which was also used to deduce the instrumental resolution.

Modelling of the data was performed in SASView (http://www.sasview.org), using the Guinier–Porod model[58]. The scattering intensity, $I(q)$, is derived from independent contributions of the Guinier form in Eq. 1,

$$I(q) = \frac{G}{Q^s} \exp\left(\frac{-q^2 R_g^2}{3-s}\right) for \ q \leq q_1 \tag{1}$$

And the Porod form, in Eq. 2

$$I(q) = \frac{D}{Q^d} for \ q \geq q_1 \tag{2}$$

where $Rg$ is the radius of gyration, $d$ is the Porod exponent, and $G$ and $D$ are the Guinier and Porod scale factors, respectively. A dimensionality parameter (3 − s) is included in the Guinier form factor to help define non-spherical objects where $s = 0$ represents spheres or globules, $s = 1$ represents cylinders or rods and $s = 2$ represents lamellae or platelets.

**Stimulated emission depletion microscopy.** Sample preparation: Fibrils formed by incubating 25 μM of WT aSyn with 10% aSynC141:AF594 in 20 mM Tris pH 7.2 at 37 °C in the dark with rotation at 200 rpm for one week. 5 μM of aSyn monomer or fibrils were incubated with purified SV (0.5 μL of 4.3 μg/μL) at a volume of 50 μL in 100 mM Tris, 100 mM KCl pH 7.2. 10 μL of sample was removed at time points of 0, 3, and 24 h. 10 μL of 100 mM Tris, 100 mM KCl pH 7.2 and were deposited on glass in an 8-well chamber (ibidi). The membrane dye mCLING-ATTO647N was added at a 1:1000 dilution (5 nM final concentration) and the sample was incubated at room temperature (RT) in the dark for 15 min. Excess sample was

removed by pipettes and the sample was fixed with 4% PFA for 10 min. The sample was washed three times with 20 mM Tris pH 7.2 before imaging. Three experimental replicates and three fields of view were imaged for STED experiments.

Stimulated emission depletion (STED) imaging was performed on a home-built pulsed STED microscope[87]. STED excitation ($\lambda$exc = 640 nm) and depletion ($\lambda$depl = 765 nm) were generated from the same titanium sapphire oscillator (Spectra-Physics Mai-Tai, Newport, Irvine, USA) operating at 765 nm. The beam was divided between two paths. In the excitation path a supercontinuum was generated by pumping a photonic crystal fibre (SCG800, NKT photonics, Cologne, Germany) and the excitation wavelength was selected by a band-pass filter (637/7 BrightLine HC, Semrock, NY, USA). Excitation and depletion pulse lengths were stretched to 56 and 100 ps, respectively, through propagation in SF66 glass and polarisation maintaining single-mode fibres. The depletion beam was spatially shaped into a vortex beam by a spatial light modulator (X10468 02, Hamamatsu Photonics, Hamamatsu City, Japan) and the beams were recombined using a short-pass dichroic mirror (FF720-SDi01, Semrock, NY, USA). Imaging was performed using a commercial point scanning microscope (Abberior Instruments, Göttingen, Germany) comprising the microscope frame (IX83, Olympus, Shinjiuku, Japan), a set of galvanometer mirrors (Quad scanner, Abberior Instruments) and a detection unit. A ×100/1.4 NA oil immersion objective (UPLSAPO 100XO, Olympus) and the Inspector software was used for data acquisition (Andreas Schönle, Max Planck Institute for Biophysical Chemistry, Göttingen, Germany). Fluorescence emission was descanned, focused onto a pinhole and detected using an avalanche photodiode (SPCM-AQRH, Excelitas Technologies, Waltham, USA). A field of view of 20 × 20 µm² and 20 nm pixel size was used.

**Correlative stimulated emission depletion microscopy – atomic force microscopy (STED-AFM).** Correlative AFM–STED imaging was performed by combining a Bioscope Resolve system (Bruker, AXS GmBH) with a custom-built STED system described above. The piezo stage of the STED microscope was removed from the inverted microscope frame, and the stage of the AFM system was used to drive both microscopes at the same time. The stage of the specific AFM system is designed so that the sample holder allows for optical detection of specimens from below, while the AFM scanning head can access the sample from above. The fields of view (FOVs) of the two microscopes were aligned so that the AFM probe was positioned in the middle of the FOV of the STED microscope, by carefully moving the AFM stage using the alignment knobs. First, the optical microscope is focussed onto the coverslip surface using low power confocal excitation to prevent photo bleaching. The confocal scan range should be set to the whole available FOV of the objective of the imaging system to make it easier to find the scanned region with the AFM system. The final, fine alignment was achieved by using a brightfield image of the "shadow" of the AFM cantilever, taken with the STED, to precisely align the AFM probe with the STED lens. STED provides a fast pre-screen of the sample for regions of interest and allows AFM to be used more efficiently in specific areas. STED images were sequentially acquired at pixel size of 50 nm, a pixel dwell time of 35 µs, STED depletion power at 180 mW, 350 µW red excitation power and 250 µW for yellow excitation, respectively. AFM images were acquired in Scanasyst mode using ScanasystFluid+ probes (Bruker), with a nominal spring constant of 0.7 N m⁻¹ and a resonant frequency of 150 kHz. Images were recorded at scan speeds of 1.5 Hz and tip–sample interaction forces between 200 and 300 pN. Large-scale images (20 µm × 20 µm) were used to register the AFM with the STED FOVs, and small (2 µm × 2 µm) scans were performed to better resolve the morphology of the fibrils. Raw AFM images were first order fitted with reference to the glass substrate using Nanoscope Analysis 9.1.

**Transmission electron microscopy.** For transmission electron microscopy (TEM) imaging of SV, 1 µL of SV (3.3 mg/mL) were incubated in 50 µL of 100 mM Tris-HCl pH 7.6, 100 mM KCl at 37 °C with and without 50 µM monomeric aSyn for 4 days without shaking. Overall, 10 µL of each sample was incubated on glow-discharged carbon-coated copper grids for 1 min before washing the grids twice with dH₂O. 2% uranyl acetate was used to negatively stain the samples for 30 s before imaging on the Tecnai G2 80-200kv TEM at the Cambridge Advanced Imaging Centre.

**Thioflavin-T based assays.** 60 µM freshly made ThT (Abcam, Cambridge, UK) was added to 50 µL of 20 µM or 60 µM aSyn per well in 20 mM Tris, 140 mM KCl, 100 nM CaCl₂, 0.05% NaN₃ pH 7.2. 0, 10, 20 or 40 nM of SV were added to wells, only compiling 0.195%, 0.39% and 0.77% of the total volume, respectively, so not to dilute the aSyn. All samples were loaded onto nonbinding, clear bottom, 96-well half area, clear plates (Greiner Bio-One GmbH, Kremsmünster, Austria). The plates were sealed with a SILVERseal aluminium microplate sealer (Grenier Bio-One GmbH). Fluorescence measurements were taken using a FLUOstar Omega plate reader (BMG LABTECH GmbH, Ortenberg, Germany). The plates were incubated at 37 °C with double orbital shaking at 200 rpm for five minutes before each read every hour for 310 h. Excitation was set at 440 nm with 20 flashes, and the ThT fluorescence intensity was measured at 480 nm emission with a 1100 gain setting. ThT assays were repeated three times using three wells for each condition. Data were normalised to the well with the maximum fluorescence intensity for each plate and the average was calculated for all experiments. A linear trend line was

fitted along the exponential phase of each ThT fluorescence curve for the 9 wells per sample condition. The trend line was used to calculate the nucleation rate, where the intercept crossed the x axis ($t_{lag}$) and the elongation rate, as the slope of the exponential phase ($k$) using Eq. (3). Data was analysed in GraphPad Prism 8.

$$y = kx - tlag \qquad (3)$$

**Determining the remaining monomer concentration of aSyn after ThT-based assays using analytical size exclusion chromatography.** To calculate the remaining aSyn monomer concentration in each well after ThT-based assays, size exclusion chromatography was performed on a high-pressure liquid chromatography (SEC) system. The contents of each well after the ThT-based assay were centrifuged at 21k × g for 20 min and the supernatant from each well was added to individual aliquots in the autosampler of the Agilent 1260 Infinity HPLC system (Agilent Technologies). 20 µL of each sample was injected onto an Advance Bio SEC column, 7.8 × 300 mm 130 Å (Agilent Technologies) in 20 mM Tris pH 7.2 at 0.8 mL/min flow-rate. The elution profile was monitored by UV absorption at 220 and 280 nm. A calibration curve of known concentrations of aSyn was used to calculate the remaining monomer concentration of aSyn in each well. Nine wells from three experiments were analysed for remaining monomer concentrations, the average value of each measurement is presented including the standard error of the mean (SEM).

**Liquid atomic force microscopy.** aSyn monomer (50 µM), with or without 10% labelled aSynC141:AF647 was incubated with or without SV (3.55 µg/µL, added 1:100 vol/vol) in 20 mM Tris, 140 mM KCl, 0.05% NaN₃, pH 7.2 for 3 days at 37 °C at 20 rpm. Fibrils were diluted to 5 µM and 20 µL was deposited onto freshly cleaved mica slides coated with 0.1% Poly-L-lysine and incubated for 30 min. Samples were washed 3x in 20 mM Tris pH 7.2, 140 mM KCl to remove non-adhered fibrils. AFM images were acquired in 20 mM Tris pH 7.2, 140 mM KCl using tapping mode on a BioScope Resolve (Bruker) with ScanAsyst-Fluid+ probes. 512 lines were acquired at a scan rate of 1.5–2 Hz per image with a field of view of 2 µm. For aSyn and SV, 25 images were acquired, for aSyn only, 13 images were acquired. Images were adjusted for contrast and exported as 2D or 3D images from NanoScope Analysis 8.2 software (Bruker).

Acquired AFM images were computationally flattened on Nanoscope Analysis 9.4 (Bruker) before import into MATLAB (MathWorks, Natick, MA, USA) using the MATLAB toolkit for Bruker. Batch analysis was performed using an in-house MATLAB script. A combination of manual and automatic segmentation of individual fibrils was performed. The height profile from each fibril was smoothed and characterised as either smooth or periodic as by a peak/trough search algorithm. For calculation of average fibril height, the mean across the whole fibril length and of the peaks were used for smooth and periodic fibrils, respectively. For periodic fibrils, the mean height of its peaks and troughs were additionally calculated.

**Coherent Raman scattering microscopy and stimulated Raman scattering microscopy.** Coherent Raman scattering microscopy (coherent anti-Stokes Raman scattering (CARS) and stimulated Raman scattering (SRS)) was performed on a Leica SP8 CARS laser scanning microscope with the additional SRS option (Leica Microsystems, Mannheim, Germany). Briefly, CARS and SRS signals were excited using two temporally and spatially overlapped pulse trains from a PicoEmerald S optical parametric oscillator (APE, Berlin, Germany). When the frequency difference between the pump ($f_p$) and the stokes ($f_s$) lasers is tuned to match exactly the vibrational frequency of a molecular bond ($f_{vib}$), then the combined action of both beams causes the resonant excitation of the vibrational mode, see Eq. 4.

$$f_p - f_s = f_{vib} \qquad (4)$$

The pump beam wavelength was fixed at 1031.25 nm, and the Stokes beam was tuneable from 750 to 940 nm, allowing the excitation of vibrations in the range of 3636 to 941 cm⁻¹. Both pulses are ~2 picoseconds in duration, providing a ~12 cm⁻¹ spectral resolution of the total system. For SRS microscopy, the Stokes beam intensity was modulated at 20 MHz using an Electro-Optical Modulator (EOM). To acquire SRS signals, the pump beam intensity was recorded in the forward direction using a silicon photodiode, and demodulated using a lock-in amplifier (Zürich Instruments, Zürich, Switzerland). CARS microscopy does not require intensity modulation, and signals are detected on photomultiplier tubes in the transmitted and epi-detected directions. Second harmonic generation (SHG)/two-photon excited fluorescence signals can be recorded simultaneously with the CARS signals. Forward SRS and epi-CARS/SHG detection are possible simultaneously. Two experiments and two fields of view were taken for SRS/CARS spectra.

**Limited proteolysis of aSyn fibrils.** 40 µM of aSyn monomer was fibrillised in the presence and absence of SV. Fibrils were incubated at 37 °C in 3.8 µg mL⁻¹ Proteinase K. 20 µL aliquots were removed at 0, 1, 5, and 15 min time points and incubated with 20 mM phenylmethylsulfonyl fluoride (PMSF) to inactivate the proteinase K. The samples were frozen and lyophilised using a LyoQuest 85 freeze-dryer (Telstar, Spain). The protein films were solubilised in Hexafluoro-2-propanol

(HFIP). HFIP was then evaporated under a stream of $N_2$ and the samples resuspended in lithium dodecyl sulphate (LDS) buffer before being heated to 100 °C, analysed by SDS-PAGE on a 4–12% Bis-Tris gel (NuPAGE, Thermo Scientific) and stained with Coomassie blue (Merck).

**Culturing and incubation of i³Neurons with aSyn**. Induced pluripotent stem cells (kindly donated by Dr Edward Avezov of UK Dementia of Research Institute) were differentiated into cortical i³Neurons and cultured following the protocol by Fernandopulle et al.[88]. Cells were tested negative for mycoplasma. 50,000 i³Neurons per well were plated and incubated for 1 h with aSyn samples at a final concentration of 500 nM. 50 µM of WT monomeric aSyn with 10% aSyn-C141:ATTO647 was incubated to aggregate in 20 mM Tris, 140 mM KCl, 0.05% NaN₃, pH 7.2 in the presence or absence of SV (1:100 v/v, 3.55 µg/µL) for 7 days with rotation at 20 rpm at 37 °C. 50 µM of WT monomeric aSyn with 10% aSynC141:ATTO647 or monomer plus SV (1:100 v/v, 3.55 µg/µL) were also prepared. The equivalent concentration to 10% ATTO657N dye only, vesicle only and vesicle and ATTO657N dye were added as controls. All samples were sonicated for 10 s at 70% amplitude prior to incubation with i³Neurons (Digital Sonifier® SLPe, model 4C15, Branson, Danbury, USA). Cells were washed and fixed with 4% PFA before imaging.

***Direct* stochastic optical reconstruction microscopy**. Cell media was removed and replaced with 4% paraformaldehyde (Merck) diluted in 1×PBS. The sample was fixed for 10 min, before changing to 1 x PBS and stored at 4 °C under dark conditions. Before imaging, 1xPBS was replaced with *direct* stochastic optical reconstruction microscopy (*d*STORM) photo-switching buffer consisting of 50 mM Tris pH 8 solution supplemented with 10 mM sodium chloride (ThermoFisher Scientific), 10% glucose (ThermoFisher Scientific), 50 mM monoethanolamine (MEA, Merck KGaA), 0.5 mg/mL glucose oxidase (Merc) and 40 µg/mL catalase (Merck). The sample was then mounted on a custom-build microscope with an IX-73 Olympus frame (Olympus) with a 647 nm laser (VFL-P-300-647-OEM1-B1, MPB Communications Inc., Quebec, Canada). Laser light entering the microscope frame was reflected from a dichroic mirror (ZT647rpc, Chroma, Bellows Fall, VT, USA) onto a 100×1.49 NA oil total internal reflection (TIRF) objective lens (UAPON100XOTIRF, Olympus), before reaching the sample. Light emitted by the sample passed through the dichroic and a set of 25 mm band-pass filters (FF01-680/42- 25, Semrock) before reaching the microscope side port. Images were then relayed onto a camera (Andor iXon Ultra 897, Oxford Instruments, Belfast, UK) by a 1.3× magnification Twincam image (Cairn, Kent, UK). The image pixel size was measured to be 117 nm using a ruled slide. Each 256 × 256 image was acquired as stacks of 15,000 images with an exposure time of 10 ms. Brightfield images were also taken to allow for calculation of soma area. Three experimental replicates and a total of 23 cells were analysed for dSTORM experiments. Fluorophore localisations are first detected using the Fiji plugin, ThunderSTORM[89], before reconstruction using an in-house MATLAB script available on request. Reconstructed images were then further analysed in a separate MATLAB script to quantify the major axis length (i.e., longest dimension) and eccentricity. Aggregate segmentation was performed by applying an intensity threshold, and aSyn uptake was normalised to soma area to account for differences in cell size.

**Statistics and reproducibility**. dSTORM images were collected for three experiments and a total of 23 cells were analysed per condition. A one-way ANOVA with Holm-Šídák tests was performed using Graph Pad Prism 8. For AFM analysis, data were collected on fibril length and periodicity for 32 aSyn only fibrils and 53 aSyn fibrils grown in the presence of SV from two experimental repeats. SRS/CARS data were acquired from two experimental samples and for two fields of view from each. Data were normalised in Origin Pro. Three experimental repeats, with three wells per condition were analysed for ThT-based aggregation assays. Data were normalised and lag time and elongations rates were calculated in Excel. An ANOVA with Brown-Forsythe and Welch tests were performed using Graph Pad Prism 8. SANS data were collected once due to restricted beamtime allocation and data collection times. Data were fitted using SASview.

**Reporting summary**. Further information on research design is available in the Nature Portfolio Reporting Summary linked to this article.

## Data availability

Raw data are available at the University of Cambridge Repository https://doi.org/10.17863/CAM.87759. SANS data acquired at the Institut Laue-Langevin, Grenoble, France are available at https://doi.org/10.5291/ILL-DATA.8-03-999. In-house MATLAB code for analysis of % fluorescence in the soma, length and eccentricity of aSyn in i³Neurons and AFM fibril length and periodicity is available on request.

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

## Acknowledgements

We would like to thank Institut Laue-Langevin for allocating us beam-time and Dr Sylvian Prévost for help preparing the samples for SANS. We would like to thank Dr Karin Müller and Dr Filomena Gallo at the Cambridge Advance Imaging Centre for help with sample preparation for TEM. ScopeM (ETH Zürich) is acknowledged for access to the Coherent Raman microscope. T. M. M. acknowledges support of the Ernest Oppenheimer Fund through provision of his Oppenheimer Research Fellowship. O.V. was supported by the Engineering and Physical Sciences Research Council (EP/L015889/1) and the Horizon Europe Program (101064246). C.W.C. was jointly funded by the Cambridge Trust and Wolfson College for her PhD. G.S.K.S. acknowledges funding from the Wellcome Trust (065807/Z/01/Z) (203249/Z/16/Z), the UK Medical Research Council (MRC) (MR/K02292X/1), Michael J Fox Foundation (16238 and 022159) and Infinitus China Ltd.

## Author contributions

A.D.S. prepared aSyn protein and SV, performed AFM, TEM, proteinase K experiments. A.F.-V. performed i³Neuronal culturing and experiments. C.W.C. performed dSTORM, and analysed dSTORM and AFM data. O.V. and E.W. performed STED microscopy. O.V. and I.M. performed correlative STED-AFM imaging. D.P. performed CARS/SRS imaging and analysed data. T.M.M. and A.D.S. analysed SANS data. R.C. ran SANS experiments. A.F.R., C.F.K. and G.S.K.S. supplied resources and scientific discussion. All authors contributed to writing the manuscript and gave their final permission.

## Competing interests

The authors declare no competing interests.
