## [Peer review file · Communications Biology]

α -synuclein fibril and synaptic vesicle interactions lead to vesicle destruction and increased lipid-associated fibril uptake into iPSC-derived neuronsReviewers' comments:

Reviewer #1 (Remarks to the Author):

In this manuscript, the authors investigate the influence of synaptic vesicles (SVs) on alpha-synuclein (aSyn). Multiple biophysical/biochemical methods have been used in this study. The paper is well-written. My main concern is that many findings have been reported, such as SV clustering by aSyn monomers, ratio-dependence of aSyn aggregation induced by lipids, and membrane damage by aSyn aggregates. The bright side of the issue is that the biological story is not off the main road.

Reviewer #2 (Remarks to the Author):

The novel aspects of these studies are:

1) the observation that fibrillar (but not monomeric) a-syn is taken up more rapidly into neurons, when incubated with synaptic vesicles (SVs); and 2) that there are interesting differences in fibrillization rates when a-syn fibrils are incubated with SVs, and the ratio of vesicles/a-syn is changed. The authors also argue that using SVs instead of traditional lipids is a strength of this study.

The experiments are well done and the data are clear, but the significance of 1) is unclear. If a-syn fibril uptake happens in vivo, the fibrils will be taken up from the extracellular space. How is incubating fibrils with SVs telling us anything about what may be going on in vivo? Obviously SVs are not present in the extracellular space. This is not a criticism of in vitro experiments, but there should at least be a conceivable way in which the in vitro experiments relate to pathophysiology. The way the experiments are presented, there is no conceivable way by which this could be relevant to what is happening in the brain, which is an issue. Instead, the authors could portray the data as evidence that incubation with lipids increases the uptake of -syn fibrils (but not monomers), without bringing in the logic of using SVs (though this is also a problem because the very design of these experiments uses SVs). Its very likely that the results are because lipids are being used in these experiments, and has nothing to do with the fact that they are SVs.

Experiments related to point 2) are interesting, though it is hard to imagine the significance of increasing the SV/protein ratio in a real world setting. These data should be extended if the authors want to have an impactful paper.

Other points:

- lines 97-98: Why exactly was the experiment done this way? "...with 500 nM aSyn monomer and fibrils labelled with 10% aSynC141-99 ATTO647N."
- Fig. 1b.i: Is M v/s M+SV statistically significant?
- line 34-36 - what does "3+" and "-12" mean? Explain if it means that those regions have a net positive or negative charge respectively.
- lines 40-41 - is ref #5 really saying that monomeric a-syn clusters a-syn? There are other papers that directly show this, besides the two cited (Diao et al., elife 2013, Sun et al., PNAS, 2019).
- 42 - Lewy neurites not Lewy Neurites. Fix language: "Lewy Bodies and Lewy Neurites do not only comprise of aSyn, but lipids, organelle membranes.."
- 79 - "Focussing..." is not a word.
- 94-95 - Fix "To use more physiologically membranes than..."
- 99-200 - Unclear: "The fibrils were either elongated in the presence and absence of SV before being sonicated to fragment..."
- 346-48 - Fix: "The mechanism 347 of which is due to a clustering of the C-terminus in the fibrillar state and thereby increasing electrostatic interactions with receptors."

Reviewer #3 (Remarks to the Author):

The present manuscript by Stephens et al. addresses an important aspect of the role a-synuclein plays in relation to synaptic vesicles, both in vivo and vitro. Lipid interactions are often overlooked in the field or more recently, specific lipids or designed lipid particles are often used for structural and functional studies. This study stands out in that it works with natively extracted synaptic vesicles and tracks a-syn behaviour in situ with microscopic methods.

I'm no expert in the latter but find the data and the interpretation convincing, providing some important insight into how fibrils can disintegrate SVs and lipids facilitate entry of a-syn. I recommend this work for publication subject to a few minor comments to address.

First, about the SVs. While they are no doubt the closest we can probably get to the in vivo setting, how far are lipids scrambled during their extraction from cells? Do they for example also contain mitochondrial lipids, which have been postulated to play a potential role in interaction with "trafficked" a-syn although the two types of membranes would usually not mix in the cell?

Maybe the authors can include a brief discussion how representative SVs are and what we know about possible differences to the lipids in situ (composition, curvature, homogeneity of the bilayer etc.).

Abstract: "(aSyn) is a well characterised lipid binding protein" - too often it is not characterised as such, lipid binding is more of an afterthought. Maybe say "well characterised, and importantly also binding lipid"?

"rather than structure." - something missing

Page 4: "To use more physiologically membranes" - rephrase

Page 6, last para "smooth aSyn fibrils (Figure 3b. i)" - this should be Fig 2b.i

Figure 1 legend, "Neuron(e)s" - I guess the spellchecker has a mind of its own, check for correct spelling everywhere

In Figure 3, I'd like the authors to double check the description where it says that "k reduces as the SV:aSyn ratio increases for both 20 μ M and 60 μ M aSyn (Figure 3b.ii)". I can see this in Fig 3b.i but not 3b.ii, if anything there is a slight increase in the latter?

Page 9, "aSyn nucleation rate, observed by the time to form fibrils, lag time (t_{lag})..." - how was that determined, time to onset of ThT increase (e.g. above 10% of baseline), or midpoint of curve?

Page 15, "increasing the SV:aSyn ratio leads to a decrease in the aSyn nucleation rate" - I wonder if this is a sort of dilution effect, rather than structural, I suggest discussing this further

It would be really interesting to find out which one of the lipids (or all) from the SVs end up coating the protein, but this would be beyond the current study.

Point-by-point response to reviewers

We thank all the reviewers for the constructive and helpful comments on our manuscript. In the following, we reply point by point to all comments raised and highlight the changes in the revised manuscript.

Reviewers' comments:

Reviewer #1 (Remarks to the Author):

In this manuscript, the authors investigate the influence of synaptic vesicles (SVs) on alpha-synuclein (aSyn). Multiple biophysical/biochemical methods have been used in this study. The paper is well-written. My main concern is that many findings have been reported, such as SV clustering by aSyn monomers, ratio-dependence of aSyn aggregation induced by lipids, and membrane damage by aSyn aggregates. The bright side of the issue is that the biological story is not off the main road.

We thank the reviewer for the comment on the well-written paper. We agree, SV clustering by aSyn monomers has been observed previously, including by our group, which we clearly stated in the manuscript. However, all the other observations raised by the reviewer have not been reported previously using physiologically relevant membranes. This view is further supported by reviewer 2, but also reviewer 3, who summarise that the novel aspects of this study as follows: 1) the use of physiological membranes that contain membrane proteins and cholesterol, which are often not present in studies using synthetic lipid membranes. 2) the increased uptake of lipid-associated aSyn fibrils into cells. Furthermore, the reviewer 2 and 3 mention that the topic of aSyn-lipid interaction is currently of high interest to the field.

Reviewer #2 (Remarks to the Author):

The novel aspects of these studies are:

1) the observation that fibrillar (but not monomeric) a-syn is taken up more rapidly into neurons, when incubated with synaptic vesicles (SVs); and 2) that there are interesting differences in fibrillization rates when a-syn fibrils are incubated with SVs, and the ratio of vesicles/a-syn is changed. The authors also argue that using SVs instead of traditional lipids is a strength of this study.

The experiments are well done and the data are clear, but the significance of 1) is unclear. If a-syn fibril uptake happens in vivo, the fibrils will be taken up from the extracellular space. How is incubating fibrils with SVs telling us anything about what may be going on in vivo? Obviously SVs are not present in the extracellular space. This is not a criticism of in vitro experiments, but there should at least be a conceivable way in which the in vitro experiments relate to pathophysiology. The way the experiments are presented, there is no conceivable way by which this could be relevant to what is happening in the brain, which is an issue. Instead, the authors could portray the data as evidence that incubation with lipids increases the uptake of -syn fibrils (but not monomers), without bringing in the logic of using SVs (though this is also a problem because the very design of these experiments uses SVs). Its very likely that the results are because lipids are being used in these experiments, and has nothing to do with the fact that they are SVs.

We thank the reviewer for their constructive criticism.

Uptake of aSyn fibrils by cells has been a field of intense investigation and there is a plethora of research articles available that support the notion that fibrillar aSyn is readily taken up by neurons and thus enhances the neuron to neuron spreading of pathology. For a review article of aSyn propagation see (Brás and Outeiro, Cells 2021). Important studies have shown that the injection of recombinant aSyn fibrils into one brain region (e.g. the substantia nigra or olfactory bulb) of mice can spread and propagate into other areas of the brain and thereby recruit endogenous aSyn (Masuda-Suzukake, et al., Brain 2013, Rey, et al., Exp. J Med 2016). It has further been shown that patients who have received human embryonic dopaminergic neuron transplants to halt the progression of the disease, contain transplanted neurons that have been infiltrated by aSyn pathology from host cells 14 years after transplantation as shown by post-mortem analysis (Kordower, et al., Nat Med, 2008). There are further reports in the literature that aSyn can be released by either exosomes, tunnelling nanotubes or by dying cells. We and others (Seetaloo, et al., Anal Chem, 2022, Munishkina, et al., Biochem. 2004) have previously shown that aSyn can readily aggregate in buffer conditions mimicking the extracellular space, which not only contains higher calcium concentrations but also a significant amount of lipids, in the form of cholesterol (cholesterol in the brain is primarily produced by astrocytes and taken up into neurons by endocytosis, similar to exogenous aSyn), but also in the form of exosomes. It is thus conceivable that aSyn can be released into the extracellular space where it encounters conditions that can influence its aggregation propensity, but also lipid structures. We thus feel our model system is valuable to investigate the role of lipids in aSyn fibril uptake.

To make it clearer that the SV serve as model physiological membranes we have amended the text as below,

Page 3: 'Here, we use SV isolated from rodent brains, as a model physiological membrane with native lipid-associated proteins and containing diverse lipids'

Page 4 'As a model physiological membrane we isolated SV from rodent brain'

Page 14 'We aimed to further our understanding of aSyn interaction with physiologically relevant membranes'

Experiments related to point 2) are interesting, though it is hard to imagine the significance of increasing the SV/protein ratio in a real world setting. These data should be extended if the authors want to have an impactful paper.

In a cellular context a change in the ratio of SV/aSyn can alter due to a few reasons, for instance, an increase of aSyn by gene duplication and triplication which increases the quantity of aSyn in cells, and which leads to early onset parkinsonism. It has further been shown that increased neuronal activity or traumatic brain injury can significantly increase the release of intracellular proteins such as aSyn (Yamada, et al., Mol Neurodegen, 2018, Acosta, et al., J Cell Physiol. 2025), which might influence the SV/protein ratio.

Other points:

- lines 97-98: Why exactly was the experiment done this way? "...with 500 nM aSyn monomer and fibrils labelled with 10% aSynC141-99 ATTO647N."

The 10% dye-labelled aSyn is optimal for imaging by dSTORM (Pinotsi et al, PNAS 2016). We have amended the text to make it clearer.

Page 5 'We treated i³Neurons with 500 nM aSyn monomer and fibrils, 10% of which was labelled with ATTO647N to permit imaging by dSTORM'

- Fig. 1b.i: Is M v/s M+SV statistically significant?

This was not statistically significant, we have amended the text to make the comparisons clearer.

Page 6 'F vs F+SV ** p<0.005, F+SV vs M and F+SV vs M+SV ****p<0.0001'

- line 34-36 - what does "3+" and "-12" mean? Explain if it means that those regions have a net positive or negative charge respectively.

This has been amended to be clearer

Page 2 'which has an overall positive charge of 3+'

Page 2 'which is highly negatively charged with an overall charge of -12,'

- lines 40-41 - is ref #5 really saying that monomeric a-syn clusters a-syn? There are other papers that directly show this, besides the two cited (Diao et al., elife 2013, Sun et al., PNAS, 2019).

References 5-7 are in reference to the release and recycling of SV, not the clustering of SV. Other references have now been added to reference the clustering of SV.

- 42 - Lewy neurites not Lewy Neurites. Fix language: "Lewy Bodies and Lewy Neurites do not only comprise of aSyn, but lipids, organelle membranes.." – this has been amended.

- 79 - "Focussing..." is not a word. – 'Focussing' can be found in the Oxford English dictionary.

- 94-95 - Fix "To use more physiologically membranes than..."

This has been amended to 'physiologically relevant membranes'

- 99-200 - Unclear: "The fibrils were either elongated in the presence and absence of SV before being sonicated to fragment...".

This has been amended to 'The fibrils were either elongated in the presence or absence of SV and the formed fibrils were consequently sonicated for fragmentation'

- 346-48 - Fix: "The mechanism 347 of which is due to a clustering of the C-terminus in the fibrillar state and thereby increasing electrostatic interactions with receptors."

This has been amended to be clearer. 'The mechanism is proposed to be via increased electrostatic interactions between receptors and fibrils compared to monomers due to a clustering of the negatively charged C-terminus in the fibrillar state'

Reviewer #3 (Remarks to the Author):

The present manuscript by Stephens et al. addresses an important aspect of the role a-synuclein plays in relation to synaptic vesicles, both in vivo and vitro. Lipid interactions are often overlooked in the field or more recently, specific lipids or designed lipid particles are often used for structural and functional studies. This study stands out in that it works with natively extracted synaptic vesicles and tracks a-syn behaviour in situ with microscopic methods.

I'm no expert in the latter but find the data and the interpretation convincing, providing some important insight into how fibrils can disintegrate SVs and lipids facilitate entry of a-syn. I recommend this work for publication subject to a few minor comments to address.

We thank the reviewer for highlighting the importance of using SV as a physiological model membrane and showing interest in aSyn-lipid species during propagation.

First, about the SVs. While they are no doubt the closest we can probably get to the in vivo setting, how far are lipids scrambled during their extraction from cells? Do they for example also contain mitochondrial lipids, which have been postulated to play a potential role in interaction with "trafficked" a-syn although the two types of membranes would usually not mix in the cell?

Maybe the authors can include a brief discussion how representative SVs are and what we know about possible differences to the lipids in situ (composition, curvature, homogeneity of the bilayer etc.).

We have added text to further describe the vesicles as below,

Page 4 'SV were used as a physiological model membrane. We isolated SV from rodent brain, using a similar protocol to those isolated and characterised in^{42,43} which contain ~36% phosphatidylcholine, 23% phosphatidylethanolamine, 19% phosphatidylinositol, 12% phosphatidylserine, 7% sphingomyelin, 3% other lipids including cholesterol, hexylceramide and ceramine and 80+ SV-associated proteins. The purified SV were imaged by TEM and ranged in sizes between 40 – 70 nm in diameter, a similar size to those identified previously⁴² but also shown to occur *in situ* as shown by electron microscopy⁴⁴ (Supplementary Figure 1).

Abstract: "(aSyn) is a well characterised lipid binding protein" - too often it is not characterised as such, lipid binding is more of an afterthought. Maybe say "well characterised, and importantly also binding lipid"?

This has been amended to 'Monomeric alpha-synuclein (aSyn) is a well characterised protein that has a high propensity to bind to lipids'

"rather than structure." - something missing

This has been removed

Page 4: "To use more physiologically membranes"

This has been amended to 'physiologically relevant membranes'

Page 6, last para "smooth aSyn fibrils (Figure 3b. i)" - this should be Fig 2b.i

This has been amended.

Figure 1 legend, "Neuron(e)s" - I guess the spellchecker has a mind of its own, check for correct spelling everywhere

Two amendments have been made.

In Figure 3, I'd like the authors to double check the description where it says that "k reduces as the SV:aSyn ratio increases for both 20 μ M and 60 μ M aSyn (Figure 3b.ii)". I can see this in Fig 3b.i but not 3b.ii, if anything there is a slight increase in the latter?

We thank the reviewer for spotting this, the graph is correct, but table 1 had 1:6000 and 1:1500 the wrong way round. This has been amended.

Page 9 'k reduces as the SV:aSyn ratio increases for 20 μ M aSyn, but is similar for 60 μ M aSyn'

Page 9, "aSyn nucleation rate, observed by the time to form fibrils, lag time (t_{lag})..." - how was that determined, time to onset of ThT increase (e.g. above 10% of baseline), or midpoint of curve?

The calculations are described in the methods section, page 24

'A linear trend line was fitted along the exponential phase of each ThT fluorescence curve for the 9 wells per sample condition. The trend line was used to calculate the nucleation rate, where the intercept crossed the x axis (t_{lag}) and the elongation rate, as the slope of the exponential phase (k) using equation 1.

$$y = kx - t_{lag} \quad (1)$$

Page 15, "increasing the SV:aSyn ratio leads to a decrease in the aSyn nucleation rate" - I wonder if this is a sort of dilution effect, rather than structural, I suggest discussing this further

The SV were purified at high concentrations, therefore the additional volume to the aSyn in the ThT-based experiments was minimal and wouldn't have 'diluted' the aSyn as such. The volume of the SV at 40 nM was 0.77%, at 20 nM 0.39% and at 10 nM 0.195% within the total volume of solution for the assay. This information has been added to the methods for clarity,

Page 14 'only compiling 0.195%, 0.39% and 0.77% of the total volume, respectively, so not to dilute aSyn concentrations'

It would be really interesting to find out which one of the lipids (or all) from the SVs end up coating the protein, but this would be beyond the current study.

We agree with the reviewer that it would be very interesting to see which lipids were bound to aSyn in different forms and whether this would make a difference to the species that could propagate and the propensity for uptake into cells. However, we agree with the reviewer, this is beyond the current study.

Reviewers' comments:

Reviewer #2 (Remarks to the Author):

I'm ok with the changes but there is one remaining issue that the authors don't seem to (or don't want to) acknowledge. In the paper the authors incubate alpha synuclein fibrils with synaptic vesicles and this is taken up more rapidly by i3 neurons. Theoretically, in the brain, the fibrils can be present either 1) inside the neuron, associated with synaptic vesicles, or 2) outside the neurons, not associated with synaptic vesicles. There is no situation one can think of where the brain would have synaptic vesicles hanging out in the extracellular space, and these "extracellular synaptic vesicles" would somehow be relevant in the transmission of alpha-synuclein fibrils.

I asked the authors how this could happen and they give a summary of the literature on cell to cell transfer, which is not the relevant point here. They end by saying "It is thus conceivable that aSyn can be released into the extracellular space where it encounters conditions that can influence its aggregation propensity, but also lipid structures." Encounters what? Encounters synaptic vesicles in the extracellular space? Are "lipid structures" the same as synaptic vesicles? Equating "lipid structures" to "synaptic vesicles" like the authors are doing diminishes their work in my opinion (and decades of work on synaptic vesicles).

My suggestion is that the authors do not highlight incubation with synaptic vesicles as a novel aspect and being any better than incubating with lipids. Perhaps the unique lipid/phospholipid composition of real synaptic membranes can be talked about, but not that the experiments are better because there are synaptic vesicles, or that incubation with synaptic vesicles can reflect the real world situation. I would suggest that instead of skirting this issue, the authors should clearly say that its not that they think that there would be extracellular synaptic vesicles, but that having a lipid composition that resembles something that could be present in the brain is an important aspect of their experiment (something like that, but written unambiguously). The way the paper is currently set up, their arguments are all treading on thin ice, and in my opinion, will turn away readers from this work in the long run.

I do think that the data that incubating with lipids increases transmission is important, and in that way the authors are making an important contribution.

Apparently the "focusing/focussing" debate goes back a long way:

<https://bridgingtheunbridgeable.com/2013/02/06/focussing-focusing/>

<https://medium.com/@wordsfromawoodward/focusing-focussing-on-the-english-language-8b7a56636c6d>

Reviewer #3 (Remarks to the Author):

It was pointed out correctly by the other reviewers that the motivation for studying a-syn fibril transfer from SVs into neurons was not very clear - I had assumed it was to study fibril spreading - but the authors have addressed this now, and I think that it also makes it much clearer what the physiological relevance of the work is. My points have been addressed well so I now recommend this paper for publication.

Point-by-point response to reviewers

We thank the reviewers for their constructive comments on our revised manuscript. In the following, we highlight the changes in the newly revised manuscript in response to reviewer 2.

Reviewers' comments:

Reviewer #2 (Remarks to the Author):

I'm ok with the changes but there is one remaining issue that the authors don't seem to (or don't want to) acknowledge. In the paper the authors incubate alpha synuclein fibrils with synaptic vesicles and this is taken up more rapidly by i3 neurons. Theoretically, in the brain, the fibrils can be present either 1) inside the neuron, associated with synaptic vesicles, or 2) outside the neurons, not associated with synaptic vesicles. There is no situation one can think of where the brain would have synaptic vesicles hanging out in the extracellular space, and these "extracellular synaptic vesicles" would somehow be relevant in the transmission of alpha-synuclein fibrils.

I asked the authors how this could happen and they give a summary of the literature on cell to cell transfer, which is not the relevant point here. They end by saying "It is thus conceivable that aSyn can be released into the extracellular space where it encounters conditions that can influence its aggregation propensity, but also lipid structures." Encounters what? Encounters synaptic vesicles in the extracellular space? Are "lipid structures" the same as synaptic vesicles? Equating "lipid structures" to "synaptic vesicles" like the authors are doing diminishes their work in my opinion (and decades of work on synaptic vesicles).

My suggestion is that the authors do not highlight incubation with synaptic vesicles as a novel aspect and being any better than incubating with lipids. Perhaps the unique lipid/phospholipid composition of real synaptic membranes can be talked about, but not that the experiments are better because there are synaptic vesicles, or that incubation with synaptic vesicles can reflect the real world situation. I would suggest that instead of skirting this issue, the authors should clearly say that its not that they think that there would be extracellular synaptic vesicles, but that having a lipid composition that resembles something that could be present in the brain is an important aspect of their experiment (something like that, but written unambiguously). The way the paper is currently set up, their arguments are all treading on thin ice, and in my opinion, will turn away readers from this work in the long run.

I do think that the data that incubating with lipids increases transmission is important, and in that way the authors are making an important contribution.

We thank the reviewer for highlighting the importance of our experiments showing that lipids increase aSyn fibril transmission. We have further amended the text (see below and green highlighted in the manuscript) to make it clear that the use of SV is not a model for SV in the extracellular space.

Fibrillar aSyn is taken up more readily by i³Neurons when exposed to physiological lipids

Within the brain, aSyn can be released and taken up by neighbouring neurons, leading to the transmission of so called aSyn seeds^{40,41}. aSyn has a very high propensity to interact with lipids through its N-terminus containing 11 imperfect repeats which become helical upon binding, similar to apolipoproteins⁴². Within insoluble inclusions, such as Lewy bodies, observed in PD patients, aSyn is found highly enriched along with lipids^{12,13}. Since both monomeric and small fibrillar aSyn can be released into the extracellular space, and due to aSyn's high affinity to lipids, it is conceivable that aSyn is associated with lipid structures when released into the extracellular space and/or bind to extracellular lipids, such extracellular vesicles, before subsequently being taken up into neighbouring neurons. Yet, there has been little study in the uptake of lipid-associated aSyn. As aSyn is primarily enriched at the presynapse where it is closely associated with SV, we used SV isolated from rodent brains^{43,44} (Supplementary Figure 1), as a model lipid system in our study. We thus compared the level of uptake into i³Neurons of aSyn monomers in the presence or absence of SV and of fibrils formed in the presence or absence of SV. Each of the above-described samples was sonicated, and the sonicated fibrils were analysed by AFM. The average length for fibrils formed in the absence of SV was 72.9 ± 45.2 nm and with SV 63.8 ± 35.1 nm, (Supplementary Figure 2).

Apparently the "focusing/focussing" debate goes back a long way:

<https://bridgingtheunbridgeable.com/2013/02/06/focussing-focusing/>

<https://medium.com/@wordsfromawoodward/focusing-focussing-on-the-english-language-8b7a56636c6d>

Perhaps we can agree to disagree?

Reviewer #3 (Remarks to the Author):

It was pointed out correctly by the other reviewers that the motivation for studying a-syn fibril transfer from SVs into neurons was not very clear - I had assumed it was to study fibril spreading - but the authors have addressed this now, and I think that it also makes it much clearer what the physiological relevance of the work is. My points have been addressed well so I now recommend this paper for publication.

We thank the reviewer for their positive comments.

REVIEWERS' COMMENTS:

Reviewer #2 (Remarks to the Author):

Looks good.